# Approximating Latent Manifolds in Neural Networks via Vanishing Ideals

**Nico Pelleriti** [1 2]  **Max Zimmer** [1 2]  **Elias Wirth** [1 2]  **Sebastian Pokutta** [1 2]

## Abstract

Deep neural networks have reshaped modern machine learning by learning powerful latent representations that often align with the manifold hypothesis: high-dimensional data lie on lower-dimensional manifolds. In this paper, we establish a connection between manifold learning and computational algebra by demonstrating how vanishing ideals can characterize the latent manifolds of deep networks. To that end, we propose a new neural architecture that (i) truncates a pretrained network at an intermediate layer, (ii) approximates each class manifold via polynomial generators of the vanishing ideal, and (iii) transforms the resulting latent space into linearly separable features through a single polynomial layer. The resulting models have significantly fewer layers than their pretrained baselines, while maintaining comparable accuracy, achieving higher throughput, and utilizing fewer parameters. Furthermore, drawing on spectral complexity analysis, we derive sharper theoretical guarantees for generalization, showing that our approach can in principle offer tighter bounds than standard deep networks. Numerical experiments confirm the effectiveness and efficiency of the proposed approach.

## 1. Introduction

Deep Neural Networks (NNs) have revolutionized various fields by learning powerful feature representations from data (Shaheen et al., 2016; Alzubaidi et al., 2021; Shi et al., 2022). A key theoretical foundation of their success is the manifold hypothesis (Cayton, 2008; Fefferman et al., 2016; Brahma et al., 2016), which suggests that high-dimensional real-world data lies on lower-dimensional manifolds, making many learning tasks tractable.

---
[1]Department for AI in Society, Science, and Technology, Zuse Institute Berlin, Germany [2]Institute of Mathematics, Technische Universität Berlin, Germany. Correspondence to: Nico Pelleriti <pelleriti@zib.de>, Max Zimmer <zimmer@zib.de>.

*Proceedings of the 42nd International Conference on Machine Learning*, Vancouver, Canada. PMLR 267, 2025. Copyright 2025 by the author(s).

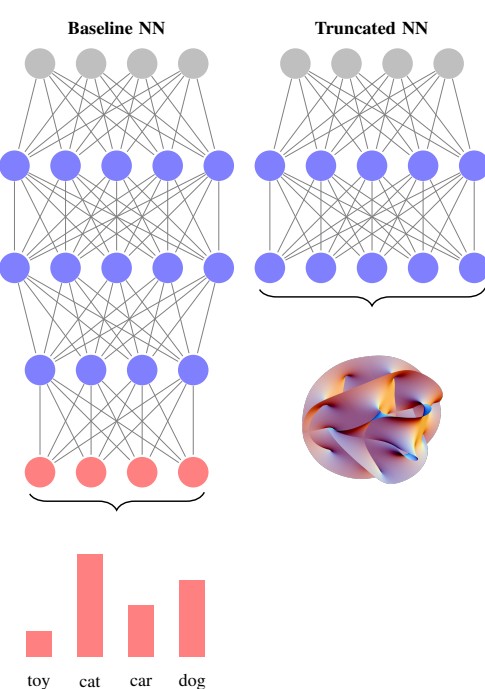

*Figure 1.* Comparison between a baseline NN (left) and a truncated NN (right). The baseline NN output is interpreted as class probabilities. The truncated NN output is interpreted using the manifold hypothesis, suggesting that classes lie on distinct manifolds. We characterize these manifolds using a set of polynomials, capturing their underlying structure. The depicted manifold is a visualization of a Calabi–Yau manifold, licensed under the Creative Commons Attribution-Share Alike 2.5 Generic license (Lunch, 2007).

Such manifolds can often be described as the zero set of a system of polynomial equations (Nash, 1952)[1]. Formally, this system gives rise to an *algebraic variety*, which is defined as

$$\mathcal{Z} = \{\, \mathbf{z} \in \mathbb{R}^n : p_i(\mathbf{z}) = 0 \text{ for all } i \in \{1, \ldots, k\}\,\},$$

for some set of polynomials $p_1, \ldots, p_k$. Given a sample $\mathbf{Z} = \{\mathbf{z}_1, \ldots, \mathbf{z}_m\} \subseteq \mathcal{Z}$ drawn from this variety, we can in turn estimate the polynomials $p_1, \ldots, p_k$ that characterize $\mathcal{Z}$ by computing the *vanishing ideal* of this sample, i.e., the

---
[1]While not every manifold is an algebraic variety, every compact smooth manifold is diffeomorphic to a real algebraic variety by the Nash-Tognoli theorem (Nash, 1952).

set of all polynomials that vanish over $\mathbf{Z}$, defined as

$$\mathcal{I}(\mathbf{Z}) = \{p \in \mathcal{P} : p(\mathbf{z}_i) = 0 \text{ for all } \mathbf{z}_i \in \mathbf{Z}\},$$

where $\mathcal{P}$ is the set of all $n$-variate polynomials. While the vanishing ideal is infinite, there exist finitely many polynomials that generate it (Cox et al., 2010). To construct such a set of *generators*, one typically employs *vanishing ideal algorithms* (Fassino, 2010; Heldt et al., 2009; Livni et al., 2013; Limbeck, 2014; Zhang, 2018; Wirth & Pokutta, 2022): For a given set of points $\mathbf{Z}$, these algorithms construct the finite set of generators, thereby characterizing $\mathcal{I}(\mathbf{Z})$. These sets of polynomials have been leveraged in downstream tasks, such as hand posture recognition, principal variety analysis for nonlinear data modeling and solution selection using genetic programming (Iraji & Chitsaz, 2017; Kera & Iba, 2016; Wang et al., 2018).

In this work, we aim to use vanishing ideal algorithms to construct polynomials that describe the *latent space* of an NN, which serves as a compressed, structured representation of the input data within the NN. Prior work (Brahma et al., 2016) has shown that this latent space often behaves like a manifold, where the network disentangles complex, high-dimensional manifolds into simpler, lower-dimensional ones.

A key observation, and central motivation for our work, is that if the latent space is a manifold, it can be algebraically characterized as the zero set of certain polynomials. Further, if the data classes correspond to disjoint algebraic varieties, then the generators of their vanishing ideals can be used as discriminative features to separate classes in the latent space, making them linearly separable (Livni et al., 2013).

In consequence, computing generators of the vanishing ideals on such latent representations is highly desirable. Unfortunately however, current vanishing ideal algorithms are incapable of processing high-dimensional data, such as images or even their representations in the latent space of NNs. While recent advancements (Wirth & Pokutta 2022) have made it feasible to compute approximate generators for data with dimensionalities up to 67, beyond these scales the computational cost grows prohibitively. Already at this scale, vanishing ideal algorithms can produce millions of polynomials, making subsequent evaluation computationally infeasible. Hence, identifying a compact and meaningful subset of generators that effectively characterizes the underlying manifold remains an open challenge.

Addressing these challenges, we introduce a set of techniques that make vanishing ideal computations feasible for high-dimensional latent spaces. Building on these advancements, we propose a novel neural network architecture, which we term Vanishing Ideal Networks (VI-Nets). Similar to linear probing (Alain & Bengio, 2016), our approach truncates a pretrained network at an intermediate

layer, but instead of just adding a linear classifier, we construct generators of the per-class vanishing ideals to capture the nonlinear structure of the latent manifolds. VI-Nets leverage these latent representations at layer $L'$ to yield a feature transformation that renders linearly-inseparable data linearly-separable, which is then used for simple discrimination and classification. Notably, while discarding all layers beyond layer $L'$, VI-Nets still achieve competitive performance while using significantly fewer parameters. We summarize our key contributions:

**1. Characterizing Manifolds in NN Latent Spaces.** Vanishing ideal algorithms require low-dimensional input points and often produce an excessive number of generators, each with numerous monomial terms needing individual evaluation. This makes their application to the high-dimensional latent space of NNs infeasible. Furthermore, the complete set of generators often fails to accurately represent the algebraic varieties of the latent space. We address these issues by: **a)** introducing dimensionality reduction and data rescaling techniques to enable the application of vanishing ideal algorithms in NN latent spaces; **b)** leveraging and adapting stochastic formulations of vanishing ideal algorithms to derive relatively sparse polynomials that can be efficiently evaluated; **c)** proposing pruning methods to significantly reduce the number of generators, preserving only those that meaningfully capture the underlying manifold.

**2. Building Efficient VI-Nets.** We propose VI-Nets, a novel NN architecture that truncates a pretrained NN and replaces its final layers with a single polynomial layer. This layer embeds the generators of each class's vanishing ideal, transforming the linearly-inseparable but polynomially-separable features of the latent space into linearly-separable ones. Unlike methods that approximate manifolds through indirect means like simplicial complexes (Khrulkov & Oseledets, 2018) or k-nearest neighbor graphs (Tsitsulin et al., 2020), our approach directly captures their algebraic structure through polynomial roots.

**3. Providing Theoretical Learning Guarantees.** We prove learning guarantees for VI-Nets by leveraging the spectral complexity framework (Bartlett et al., 2017). Specifically, we exploit the sparsity of the constructed generators to derive better generalization error bounds when comparing to the corresponding pretrained NN.

**4. Training VI-Nets.** We perform extensive experiments that showcase that VI-Nets can achieve performance competitive to the pretrained baseline NNs while using much fewer parameters. In particular, we analyze the trade-off between model accuracy and parameter count, when truncating more and more layers from the pretrained NN.

To the best of our knowledge, leveraging tools from Computational Algebra in the context of Deep Learning is a relatively underexplored field. By demonstrating the feasibility of using vanishing ideal computations to characterize data manifolds in NN latent spaces, we aim to encourage further research in this direction.

# 2. Algebraic Characterization of Data Manifolds

In this section, we present our methodology for approximating data manifolds using generators of the vanishing ideal, specifically addressing the challenges of applying vanishing ideal algorithms to NN latent spaces. Our approach constructs class-specific polynomial sets that characterize the underlying manifold structure, which we then leverage to build efficient polynomial layers. We begin by formally stating our core problem:

**Problem 2.1.** For each input class $k \in [K]$, let $\mathbf{Z}^k = \{\mathbf{z}_1^k, \ldots, \mathbf{z}_m^k\} \subseteq \mathbb{R}^n$ represent the latent space activations at some layer $L'$ of an NN for samples belonging to class $k$. The goal is to compute a subset of generators of the approximate vanishing ideal of $\mathbf{Z}^k$ that effectively describe the underlying manifold $U^k$.

## 2.1. Preliminaries

Let $l, k, m, n, q, K \in \mathbb{N}$. Vectors and matrices are denoted in **bold**. Define $\mathcal{T}$ and $\mathcal{P}$ as the sets of monomials and polynomials, respectively. A polynomial $g \in \mathcal{P}$ is expressed as $g = \sum_{i=1}^{q} c_i t_i$, with coefficients $\mathbf{c} = (c_1, \ldots, c_q)$ and monomials $t_i \in \mathcal{T}$ for $i \in [q]$. We focus on the latent representations of NNs, specifically the output activations of a specific layer $L'$. These vectors, denoted as $\mathbf{z} \in \mathbb{R}^n$, are derived from passing a sample $\mathbf{x}$ through an NN $\phi$. We focus on NNs for classification tasks. For notational simplicity, functions are often applied to sets, e.g., $\phi(\mathbf{X}) = \{\phi(\mathbf{x}) : \mathbf{x} \in \mathbf{X}\}$ for a set $\mathbf{X}$. Superscript indices denote class membership, while subscript indices denote specific elements in a set, e.g., $\mathbf{x}_1^k, \mathbf{x}_2^k, \mathbf{x}_3^k$ are three distinct input samples (interpreted as vectors) of class $k$.

Given some set $\mathbf{Z} = \{\mathbf{z}_1, \ldots, \mathbf{z}_m\} \subseteq \mathbb{R}^n$, we follow Cox et al. (2010) and define the *vanishing ideal* as follows.

**Definition 2.2.** Given sample $\mathbf{Z}$, the *vanishing ideal* $\mathcal{I}(\mathbf{Z}) \subseteq \mathcal{P}$ is defined as

$$\mathcal{I}(\mathbf{Z}) = \{g \in \mathcal{P} : g(\mathbf{z}_i) = 0 \text{ for all } i \in [m]\}.$$

Conversely, given a set of polynomials $\mathcal{F} \subseteq \mathcal{P}$, the corresponding *algebraic variety* $\mathcal{V}(\mathcal{F}) \subseteq \mathbb{R}^n$ is defined as

$$\mathcal{V}(\mathcal{F}) = \{\mathbf{z} \in \mathbb{R}^n : g(\mathbf{z}) = 0 \text{ for all } g \in \mathcal{F}\}.$$

Despite $\mathcal{I}(\mathbf{Z})$ being an infinite set, there exist finitely many polynomials that generate it (Cox et al., 2010). Further, such

generators can be computed using a variety of algorithms (Livni et al., 2013; Limbeck, 2014; Wirth & Pokutta, 2022).

## 2.2. Class Separation in the Latent Space

A key insight is that data points from different classes in the latent space, while not linearly separable, can be distinguished through polynomial separability. This concept is central to VI-Nets, which we will define later. Consider a pretrained NN $\phi$ that we modify by truncating a certain number of its final layers, resulting in $\hat{\phi}$ (see Figure 1). Let $\mathbf{X}^k$ represent input data points for class $k \in [K]$, such as those from a validation dataset. Correspondingly, $\mathbf{Z}^k$ denotes the latent space representations after truncation, i.e., $\mathbf{Z}^k = \hat{\phi}(\mathbf{X}^k)$. For each class $k$, we anticipate that the set of output activations $\mathbf{Z}^k$ resides on a manifold that can be approximated using generators of the vanishing ideal of $\mathbf{Z}^k$.

Suppose now that for each class $k$, we are given $n_k$ generators $p_1^k, \ldots, p_{n_k}^k$ of the vanishing ideal of $\mathbf{Z}^k$. Then, we define the feature transformation for a latent space sample $\mathbf{z} \in \mathbb{R}^n$ as:

$$g^k(\mathbf{z}) = \left(|p_1^k(\mathbf{z})|, \ldots, |p_{n_k}^k(\mathbf{z})|\right) \in \mathbb{R}^{n_k}.$$

Aggregating these feature transformations for all classes yields the mapping:

$$G(\mathbf{z}) = \left(g^1(\mathbf{z}), \ldots, g^K(\mathbf{z})\right) \in \mathbb{R}^{n_1 + \cdots + n_K}. \quad \text{(POLY)}$$

Typically, the latent space subsets $\mathbf{Z}^k$ are not linearly separable. While additional non-linear layers might achieve linear separability, this is not assured at intermediate layers. By definition of the vanishing ideal, the generators $p_1^k, \ldots, p_{n_k}^k$ for class $k$ vanish on $\mathbf{Z}^k$ but most likely not on $\mathbf{Z}^i$ for $i \neq k$. Thus, for a point $\mathbf{z} \in \mathbf{Z}^k$, $G(\mathbf{z})$ is zero at coordinates for class $k$ generators and positive elsewhere, enabling a linear classifier to separate the classes. For an input $\mathbf{x}$, the mapping

$$\mathbf{x} \mapsto \mathbf{W} G(\hat{\phi}(\mathbf{x}))$$

allows linear classification, where $\mathbf{W} \in \mathbb{R}^{K \times (n_1 + \cdots + n_K)}$ is the weight matrix mapping to the $K$ class logits.

## 2.3. Practical Data Manifold Approximation

Approximating class data manifolds with algebraic varieties provides a structured method to capture data relationships. As we have shown, this approach can further be used to construct a classifier from a truncated NN. However, real-world data is noisy, making it impractical to find polynomials that vanish exactly on the data. Furthermore, we argue that computing the *exact* vanishing ideal of a sample $\mathbf{Z}$ is not only computationally intractable but also lacks generalization properties. Since $\mathbf{Z}$ is finite, it itself forms a variety (e.g. as

the roots of an appropriate polynomial), and by the ideal-variety correspondence (Cox et al., 2010), $\mathcal{V}(\mathcal{I}(\mathbf{Z})) = \mathbf{Z}$. Thus, the algebraic variety defined by the vanishing ideal's generators includes only the points in $\mathbf{Z}$, offering no generalization beyond the sample. However, for class $k$, our aim is to approximate the underlying manifold $U^k$ from which $\mathbf{Z}^k$ is drawn, not just to describe $\mathbf{Z}^k$. In general, recovering $U^k$ exactly from a finite sample is impossible without structural information, as $U^k$ could range from being as minimal as $\mathbf{Z}^k$ to as broad as $\mathbb{R}^n$.

Hence, research has mainly focused on constructing approximate generators for the vanishing ideal, i.e., polynomials that $\psi$-*approximately vanish* over the data allowing a small error tolerance $\psi \geq 0$ (Heldt et al., 2009; Livni et al., 2013; Limbeck, 2014; Wirth & Pokutta, 2022). We define $\psi$-*approximately vanishing* polynomials as follows:

**Definition 2.3.** For $\psi \geq 0$, a polynomial $g = \sum_{i=1}^{k} c_i t_i$ is $\psi$-*approximately vanishing* over $\mathbf{Z}$ if $\mathrm{MSE}(g, \mathbf{Z}) = \frac{1}{m} \sum_{i=1}^{m} g(\mathbf{z}_i)^2 \leq \psi$ and its leading coefficient equals 1.

Setting the leading coefficient (i.e., the highest degree monomial's coefficient) to 1 is essential for defining approximately vanishing polynomials. Without this constraint, any polynomial $g \in \mathcal{P}$ can be rescaled to approximately vanish for all $\mathbf{z} \in \mathbf{Z}$, causing the *spurious vanishing problem* (Kera & Hasegawa, 2019), where polynomials fail to capture meaningful data structure. Unfortunately, the ideal generated by all $\psi$-approximately vanishing polynomials, known as the *approximate vanishing ideal*, also fails to provide the desired structure (Heldt et al., 2009), as it typically becomes the unit ideal, encompassing the entire polynomial ring.

To summarize: In contrast to the *exact* vanishing ideal, which is too restrictive and only captures the points in $\mathbf{Z}$, the *approximate* vanishing ideal generates the entire polynomial ring and does not depend on $\mathbf{Z}$ at all. Thus, we focus on identifying a subset $\mathcal{G}$ of its constructed generators, approximating the underlying variety and therefore manifold effectively by $\mathcal{V}(\mathcal{G})$.

## 3. Constructing practical VI-Nets

We introduced the core concept of VI-Nets, using generators of the vanishing ideal to approximate per-class data manifolds. Now, we address the identified challenges: We start with our approach for finding polynomial generators that meaningfully describe the underlying manifolds, followed by adaptations of existing vanishing ideal algorithms to being able to compute the generators in the first place.

### 3.1. Selecting Meaningful Polynomials

We derived that computing the exact vanishing ideal is too restrictive, only capturing points in $\mathbf{Z}$, while the approximate vanishing ideal generates the entire polynomial ring. To meaningfully approximate the underlying manifold $U_k$ for each class $k \in [K]$, we select a *subset* of the constructed generators of the approximate vanishing ideal with some tolerance $\psi > 0$, based on two key criteria: **a)** *low complexity* and **b)** *discriminative power*. We detail these approaches below and provide theoretical justification in Section 4 from the perspective of statistical learning theory.

**Low Complexity: Finding coefficient-sparse generators.** The number of monomial terms increases exponentially with the polynomial's maximal degree, making evaluation inefficient. A polynomial's complexity is determined by the sparsity of its coefficient vector $\mathbf{c}$; greater sparsity implies fewer non-zero coefficients and thus fewer monomial terms to compute. To achieve this, we use the meta-algorithm Oracle Approximate Vanishing Ideal Algorithm (OAVI) (Wirth et al., 2023; Wirth & Pokutta, 2022) for vanishing ideal generation, adopting the Frank-Wolfe algorithm (Frank & Wolfe, 1956; Jaggi, 2013) as its oracle to produce sparse coefficients. We also employ the Approximate Buchberger-Möller Algorithm (ABM) (Limbeck, 2014), which similarly yields sparse generators (Wirth & Pokutta, 2022), and limit the maximal degree to avoid overfitting. While both OAVI and ABM yield coefficient-sparse polynomials, they can still generate many that do not meaningfully approximate $U^k$, an issue addressed next.

**Discriminative Power: Finding distinctive generators.** We ensure selecting polynomials that effectively differentiate $U^k$ from other class regions by eliminating those that also vanish for other classes. Our experiments demonstrate that most generators capture the dataset's global structure rather than class-specific features, so we introduce a pruning score to rank each generator based on its non-vanishing ability on other classes. Let $\mathbf{Z}^k = \{\mathbf{z}_1^k, \ldots, \mathbf{z}_{m_k}^k\}$ represent the latent representations for class $k$. For a generator $p$ associated with class $j$, its score $s^j(p)$ is defined as

$$s^j(p) = \min_{k \neq j} \frac{1}{m_k} \sum_{i=1}^{m_k} \left| p(\mathbf{z}_i^k) \right|.$$

This score measures the minimal average vanishing of generator $p$ for class $j$ on other classes. Low scores indicate that a generator nearly vanishes on some class $k$, despite being a generator for class $i$, showing its lack of distinctiveness. We rank generators using $s^j(g)$ and select the top $p \%$ for each of the class $j$, as the manifolds for different classes likely exhibit varying complexity (Brown et al., 2023). The remaining $N$ generators are the most discriminative and are used to form the polynomial layer, while others are discarded. In Section 5, we validate our method, demonstrating that a subset of generators suffices for classification.

Evaluating many sparse polynomials can still be compu-

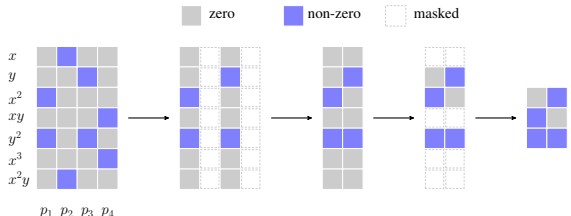

*Figure 2.* Illustration of our method for obtaining parameter-efficient representations of the most discriminatory vanishing ideal generators. The process begins by pruning columns corresponding to polynomials with the lowest pruning scores, such as $p_2$ and $p_4$ in this example. Exploiting the sparsity of the generated polynomials, rows with only zero entries—indicating monomials that do not contribute—do not need to be evaluated. As a result, only the essential monomials ($y$, $x^2$ and $y^2$ in this case) need to be evaluated during the final computation. Experiments confirm that this approach effectively reduces the number of evaluated monomials (cf. Figure 6 in the appendix).

tationally intensive due to the lack of sparsity overlap in their coefficient vectors: different polynomials might have entirely different monomials. Our two-fold approach addresses this: First, we use OAVI or ABM to generate sparse coefficient vectors. Second, we prune the set of generators to enhance discriminative power, significantly reducing the number of polynomials and indirectly also the number of monomial terms that need to be evaluated, allowing us to drastically shrink the matrix of coefficients. The remaining coefficients are then fine-tuned via gradient descent to recover any lost accuracy. This process, visualized in Figure 2, focuses on identifying a subset of generators that effectively approximate the data manifold.

### 3.2. Adapting Vanishing Ideal Algorithms to the Latent Space

In the previous section, our pruning approach assumed access to the full set of generators for each class's vanishing ideal. While ABM and OAVI (both fully described in Appendix C) are known for constructing sparse polynomials (Wirth & Pokutta, 2022), directly applying these methods to high-dimensional latent spaces with large sample sizes is computationally infeasible. Recent advances demonstrate their effectiveness on moderate-dimensional datasets (Wirth et al., 2023), but making them tractable in large-scale settings requires further adaptation.

We propose three key adaptations. First, we adopt *stochastic* variants of OAVI or ABM, enforcing vanishing on random subsets of each class' rather than on the entire class dataset. This significantly reduces computational overhead and memory usage. Second, we perform *dimensionality reduction* (e.g., via PCA) in the latent space, effectively reducing the dimensionality of input features of OAVI and ABM and

thereby controlling the growth of monomial terms. Third, we employ *lower-precision arithmetic* (e.g., float16) in the vanishing ideal computations and a *tanh-based rescaling* (Hampel et al., 2011) (cf. Appendix C.4) of latent features to stabilize numerical computations. Although these choices can slightly reduce initial accuracy, they preserve the crucial monomial structure, which we later refine by *retraining* coefficients via gradient descent. This pipeline thus retains the principal strengths of OAVI and ABM—namely, the ability to discover sparse, class-specific polynomials—while ensuring improved scalability in the setting of latent spaces within neural networks.

### 3.3. The VI-Net Pipeline

Algorithm 1 describes the VI-Net pipeline. It starts by training or using a pretrained network $\phi$, truncating it at layer $L'$, and extracting features $\mathbf{Z}^k$ for each class $k \in [K]$. Generators $p_1^k, \ldots, p_{n_k}^k$ of the vanishing ideal of $\mathbf{Z}^k$ are computed using stochastic variants of OAVI and ABM, followed by pruning. The pruned generators are aggregated to construct the transformation $G$ from (POLY). Finally, the coefficients of the polynomials in $G$ and the weights $\mathbf{W}_F$ of the appended linear layer are jointly retrained to optimize classification performance.

---

**Algorithm 1** VI-Net Pipeline

---

1: **Input:** Dataset $\mathbf{X} = \{\mathbf{x}_i^k\}_{i,k}$, NN $\phi$, intermediate layer $L'$, pruning function PRUNE, vanishing ideal algorithm VANISH.
2: **Output:** Trained VI-Net $\tilde{\phi}_{L'}$.
3: $\phi \leftarrow$ TRAIN$(\phi, \mathbf{X})$
4: $\phi_{L'} \leftarrow$ TRUNCATE$(\phi, L')$
5: **for** class $k \in [K]$ **do**
6:     $\mathbf{Z}^k \leftarrow \phi_{L'}(\mathbf{X}^k)$
7:     $p_1^k, \ldots, p_{n_k}^k \leftarrow$ VANISH$(\mathbf{Z}^k)$
8: **end for**
9: **for** class $k \in [K]$ **do**
10:     $p_1^k, \ldots, p_{n_k'}^k \leftarrow$ PRUNE$((p_1^k, \ldots, p_{n_k}^k), \mathbf{Z})$
11:     $g^k \leftarrow (|p_1^k|, \ldots, |p_{n_k'}^k|)$
12: **end for**
13: $G \leftarrow (g^1, \ldots, g^K)$, $\mathbf{W}_F \leftarrow$ INIT$(K, n_1' + \ldots + n_K')$
14: $\mathbf{W}_F \leftarrow$ TRAIN$(\mathbf{W}_F, G(\mathbf{Z}))$
15: $\tilde{\Phi}_{L'} \leftarrow \mathbf{W}_F \circ G \circ \Phi_{L'}$

---

## 4. Theoretical Analysis

We establish learning guarantees for VI-Nets, leveraging the sparsity of the coefficient matrix obtained through pruning (cf. Figure 2). We demonstrate that VI-Nets can achieve lower *spectral complexity* (Bartlett et al., 2017) than baseline NNs, resulting in a tighter bound on the generalization error.

### 4.1. Spectral Complexity of VI-Net

Following Bartlett et al. (2017), we formally introduce the spectral complexity of an NN. To that end, let $\|\mathbf{W}\|_2$ and $\|\mathbf{W}\|_{p,q} := \|(\|\mathbf{W}_{:,1}\|_p, \ldots, \|\mathbf{W}_{:,m}\|_p)\|_q$ denote the spectral norm and the $(p,q)$-norm of matrix $\mathbf{W}$, respectively.

**Definition 4.1** (Spectral Complexity). Let $\phi$ be an NN of depth $L$ with $\rho_i$-Lipschitz activation functions $\{\sigma_i\}_{i=1}^L$ and weight matrices $\{\mathbf{W}_i\}_{i=1}^L$, i.e., $\phi$ is given as the function $\phi \equiv \sigma_L \circ \mathbf{W}_L \circ \sigma_{L-1} \circ \mathbf{W}_{L-1} \circ \cdots \circ \sigma_1 \circ \mathbf{W}_1$. The spectral complexity $R_\phi$ of $\phi$ is defined as

$$R_\phi := \left( \prod_{i=1}^L \rho_i \|\mathbf{W}_i\|_2 \right) \left( \sum_{i=1}^L \frac{\|\mathbf{W}_i^T\|_{2,1}^{2/3}}{\|\mathbf{W}_i\|_2^{2/3}} \right)^{3/2}.$$

In the following, we will show that we can encode VI-Nets in the format of Definition 4.1, which allows us to provide learning guarantees, since by Bartlett et al. (2017, Theorem 1.1), the generalization error of $\phi$ is proportional to its spectral complexity $R_\phi$. To that end, we require the following assumption:

**Assumption 4.2.** Let the polynomial layer $G : \mathbb{R}^n \to \mathbb{R}^N$ be as in (POLY), where $N := n_1 + \cdots + n_K$. Let further $S \in \mathbb{N}$ be the number of monomial terms needed to evaluate $G$ and $d \in \mathbb{N}$ the highest degree of any polynomial in $G$. We assume that there exists $\tau > 0$ such that $\|\mathbf{c}_i^k\|_1 \leq \tau$ i.e., the coefficient vectors $\mathbf{c}_i^k$ of each polynomial $p_i^k$ are bounded in the $\ell_1$-norm by $\tau$.

This assumption is justified as bounded coefficient vectors can be ensured either directly by the vanishing ideal algorithm (e.g., OAVI, where $\tau$ is a hyperparameter) or indirectly through our pruning and fine-tuning approach, which adjusts and sparsifies the polynomials post hoc. Next, we introduce an activation function, that allows us to encode the polynomial layer in the format of Definition 4.1.

**Definition 4.3** (InEx-activation function). Let $d \in \mathbb{N}$ as in Assumption 4.2. Abbreviating $j_k := \sum_{i=1}^k \binom{d}{i}$, the *InEx-activation function* of degree $d$ is defined as

$$\alpha^d : \mathbb{R}^{2^d} \to \mathbb{R} \quad \mathbf{z} \mapsto \frac{1}{d!} \sum_{k=1}^d (-1)^{d-k} \sum_{j=j_{k-1}}^{j_k} (z_j)^d.$$

The following lemma leverages that the InEx-activation function, based on the *inclusion-exclusion principle*, computes monomial terms by selecting the appropriate variables using the rows in the matrix $\mathbf{W}_M$. The vector $\sigma(\mathbf{W}\mathbf{x})$ thus represents the $S$ monomial terms needed for evaluating $G$. These terms are then multiplied by the coefficient matrix $\mathbf{W}_C$ to complete the evaluation.

**Lemma 4.4.** *Let $G$ and $S$ as in Assumption 4.2 and define the stacked* InEx*-activation function $\sigma = (\alpha^{d_1}, \ldots, \alpha^{d_S})$ :*

$\mathbb{R}^s \to \mathbb{R}^S$ *for $s \in \mathbb{N}$. Then, $G$ can be represented as the two layer NN $[\mathbf{W}_C \circ \sigma \circ \mathbf{W}_M]$, where $\mathbf{W}_M \in \mathbb{R}^{s \times n}$ and $\mathbf{W}_C \in \mathbb{R}^{N \times S}$ such that $s \in \mathcal{O}(S)$.*

The entire VI-Net consists of the $L'$ remaining truncated layers of the NN, the matrices $\mathbf{W}_M, \mathbf{W}_C$ from Lemma 4.4 and a final, linear layer $\mathbf{W}_F$. The spectral complexity of $\tilde{\phi}_{L'}$ can hence be computed analogously to Definition 4.1, albeit with $L' + 3$ layers instead of $L$.

### 4.2. Learning Guarantees for VI-Nets

We now analyze the spectral properties of matrices $\mathbf{W}_M$ and $\mathbf{W}_C$ to establish an upper bound on the spectral complexity of the VI-Net, enabling the spectral complexity comparison with the baseline NN. Consider an NN $\phi$ with $L$ layers, defined by weight matrices $\{\mathbf{W}_i\}_{i=1}^L$ and $\rho_i$-Lipschitz activation functions $\{\sigma_i\}_{i=1}^L$, as per Definition 4.1. Let $\phi_{L'} \equiv \sigma_{L'} \circ \mathbf{W}_{L'} \circ \sigma_{L'-1} \circ \mathbf{W}_{L'-1} \circ \cdots \circ \sigma_1 \circ \mathbf{W}_1$ represent the truncated NN of depth $L' < L$. The corresponding encoded VI-Net $\tilde{\phi}_{L'} \equiv \mathbf{W}_C \circ \sigma \circ \mathbf{W}_M \circ \phi_{L'}$ has $L' + 3$ layers and follows the format of Definition 4.1, incorporating the polynomial layer $G$ from (POLY), with $G$ satisfying Assumption 4.2.

To simplify notation, we denote the weight matrices of $\tilde{\phi}_{L'}$ as $(\tilde{\mathbf{W}}_1, \ldots, \tilde{\mathbf{W}}_{L'+3})$ with corresponding Lipschitz constants $(\tilde{\rho}_1, \ldots, \tilde{\rho}_{L'+3})$. Note that $\tilde{\mathbf{W}}_i = \mathbf{W}_i$ for $i \in \{1, \ldots, L'\}$ and $\left( \tilde{\mathbf{W}}_{L'+1}, \tilde{\mathbf{W}}_{L'+2}, \tilde{\mathbf{W}}_{L'+3} \right) = (\mathbf{W}_M, \mathbf{W}_C, \mathbf{W}_F)$, analogously for the Lipschitz constants, however noting that for the new layers, we have $(\tilde{\rho}_{L'+1}, \tilde{\rho}_{L'+2}, \tilde{\rho}_{L'+3}) = (d, 1, 1)$.

For the remainder, we assume that the output of the truncated network $\tilde{\phi}_{L'}$ is already restricted to the hypercube $[-1, 1]^n$. In practice, this is achieved by appending a final $\tanh$ activation, which also enhances numerical stability (see Appendix C.4).

**Theorem 4.5.** *Let the networks $\phi, \phi_{L'}$ and $\tilde{\phi}_{L'}$ as above. Then, the spectral complexities of $\phi$ and $\tilde{\phi}_{L'}$ can be related as $R_{\tilde{\phi}_{L'}} = \kappa \cdot R_\phi$, where $\kappa$ is given by*

$$\kappa = \frac{\prod_{i=L'+1}^{L'+3} \tilde{\rho}_i \left\| \tilde{\mathbf{W}}_i \right\|_2}{\prod_{i=L'+1}^{L} \rho_i \left\| \mathbf{W}_i \right\|_2} \left( \frac{\sum_{i=1}^{L'+3} \left\| \tilde{\mathbf{W}}_i^T \right\|_{2,1}^{\frac{2}{3}} \left\| \tilde{\mathbf{W}}_i \right\|_2^{-\frac{2}{3}}}{\sum_{i=1}^{L} \left\| \mathbf{W}_i^T \right\|_{2,1}^{\frac{2}{3}} \left\| \mathbf{W}_i \right\|_2^{-\frac{2}{3}}} \right)^{\frac{3}{2}}.$$

*If there further exist $\lambda_1, \lambda_2 > 0$, such that $\left\| \tilde{\mathbf{W}}_{L'+3} \right\|_2 \leq \lambda_1$*

*and $\left\| \tilde{\mathbf{W}}_{L'+3} \right\|_{2,1} \leq \lambda_2 \left\| \tilde{\mathbf{W}}_{L'+3} \right\|_2$, then it holds that*

$$\prod_{i=L'+1}^{L'+3} \left\| \tilde{\mathbf{W}}_i \right\|_2 \leq 2^d d\tau \lambda_1 \sqrt{NS},$$

$$\sum_{i=L'+1}^{L'+3} \frac{\left\| \tilde{\mathbf{W}}_i^T \right\|_{2,1}^{\frac{2}{3}}}{\left\| \tilde{\mathbf{W}}_i \right\|_2^{\frac{2}{3}}} \leq 2^{\frac{2d}{3}} S^{\frac{2}{3}} + N^{\frac{2}{3}} S^{\frac{1}{3}} + \lambda_2^{\frac{2}{3}}.$$

This result provides a theoretical way to set the hyperparameters of our VI-Net. Given access to the weights of the baseline network, we can compute the spectral norms of its layers using power iteration or singular value decomposition. These norms, together with the theorem's bounds, allow us to determine appropriate values for the degree $d$ of the polynomials, the number of generators $N$, the coefficient bound $\tau$ and the number of monomials $S$ to ensure the VI-Net has lower spectral complexity.

As a corollary, we derive a bound on the generalization error of a VI-Net, leveraging the spectral complexity (Theorem 4.5) and the margin-based generalization framework of Bartlett et al. (2017). The margin measures the gap between the network's output logit for the correct label and the closest output for any incorrect label, i.e., $\tilde{\phi}_{L'}(\mathbf{x})_y - \max_{k \neq y} \tilde{\phi}_{L'}(\mathbf{x})_k$.

**Corollary 4.6.** *Let $\tilde{\phi}_{L'}$ be a VI-Net of width $\omega$ and let $R_{\tilde{\phi}_{L'}}$ be given as in Theorem 4.5. Then, for $(x, y), (x_1, y_1), \ldots, (x_m, y_m)$ drawn i.i.d. from any probability distribution over $\mathbb{R}^d \times [K]$, we have with probability at least $1 - \delta$ over $((x_i, y_i))_{i=1}^m$, every margin $\gamma > 0$ satisfies*

$$\mathbb{P}\left[ \arg\max_k \tilde{\phi}_{L'}(x)_k \neq y \right]$$
$$\leq \mathcal{L}_\gamma(\tilde{\phi}_{L'}) + O\left( \frac{\|X\|_{2,2} R_{\tilde{\phi}_{L'}}}{\gamma \sqrt{m}} \ln(\omega) + \sqrt{\frac{\ln(1/\delta)}{m}} \right),$$

*where*

$$\mathcal{L}_\gamma(\tilde{\phi}_{L'}) = \frac{1}{m} \sum_{i=1}^m \mathbb{I}\left[ \tilde{\phi}_{L'}(x_i)_{y_i} - \max_{k \neq y_i} \tilde{\phi}_{L'}(x_i)_k \leq \gamma \right],$$

$\mathbb{I}$ *the indicator function, and $X := (x_1, \ldots, x_m) \in \mathbb{R}^{d \times m}$.*

## 5. Numerical Experiments

[2]We assess the capability of VI-Net to replace entire convolutional and linear layers in standard CNNs, aiming to maintain or enhance accuracy while improving parameter

---

[2]Our code is available at https://github.com/ZIB-IOL/approximating-neural-network-manifolds

efficiency. Our experiments use a pretrained baseline neural network, leveraging only its latent outputs, and apply approximate vanishing ideal computations to these features. The primary goal is to determine if the network's final layers can be replaced with a polynomial layer from vanishing ideal generators, without retraining the entire model.

Specifically, we **(i)** demonstrate that VI-Net achieves competitive accuracy, and **(ii)** examine the impact of network truncation and polynomial pruning on accuracy, parameter efficiency, and throughput (measured in images per second). Inspired by recent findings that deeper layers in large language models are often less effective than earlier ones (Gromov et al., 2025), we intend to replace deeper layers with a polynomial layer for a more compact yet equally expressive model.

### 5.1. General Experimental Setup

All experiments use PyTorch (Paszke et al., 2019). We evaluate classification on CIFAR-10/-100 (Krizhevsky, 2009) with ResNet models (He et al., 2015). VI-Net replaces the final layers of a pretrained network with polynomial generators (Algorithm 1), where we use the following parameters: vanishing tolerance $\psi = 0.1$ (cf. Definition 2.3), maximal polynomial degree $d = 5$, and feature reduction to 128 dimensions using PCA. Fine-tuning of the linear classifier and coefficient matrix uses SGD (learning rate 0.05, momentum 0.9) with standard data augmentations.

### 5.2. Impact of Truncation

We first analyze the impact of removing varying fractions of the final convolutional layers on accuracy and the number of constructed generators for CIFAR-100, with CIFAR-10 results deferred to Appendix A.2.1. We compare three setups, all truncating the network at the same layer but differing in the layers they append and retrain: **i)** VI-Net, which appends the polynomial layer followed by a linear classifier, **ii)** *Linear Head*, which appends only a linear classifier to the truncated network, and **iii)** *Random Monomials*, which appends a polynomial layer with the same number of monomials and an identically shaped coefficient matrix as VI-Net, but with randomly sampled monomials instead of those found by the vanishing ideal algorithms.

**The effect of truncation on performance.** Figure 3 shows how truncating ResNet-34 layers affects CIFAR-100 test accuracy. Removing layers and adding only a linear layer reduces network performance due to non-linearly separable latent features. In contrast, VI-Net maintains competitive accuracy even after removing half of the layers. It consistently outperforms the random monomials layer, highlighting that VI-Net's strength lies not just in the non-linearity of the monomials but in their discriminative power

when determined by the vanishing ideal algorithms. At the start of the residual blocks, VI-Net and the monomials layer show similar accuracy, likely because more monomials are generated in this region (cf. Appendix A.2.2).

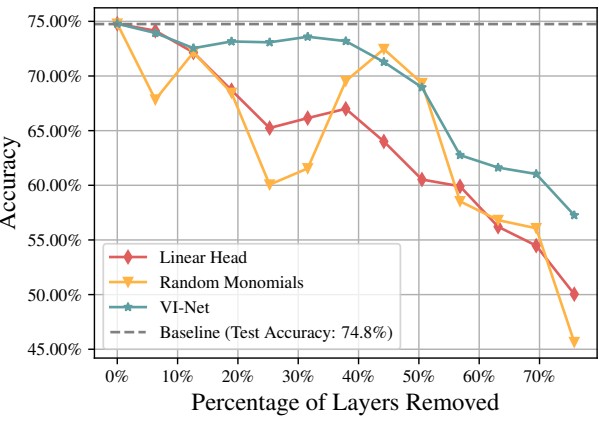

*Figure 3.* CIFAR-100 (ResNet-34): Test accuracy vs. percentage of removed convolutional layers. As more layers are removed, a classifier with only a linear head after truncation experiences a sharp performance drop, whereas VI-Net maintains accuracy.

**The effect of truncation on the generators.** Table 1 shows the number of monomials and polynomials ABM constructed after removing different fractions of layers in ResNet-18 and ResNet-34. Truncating earlier layers increases the *intrinsic dimensionality* (Ansuini et al., 2019) of data manifolds, resulting in more monomial terms (cf. Appendix A.2.8). Despite significant truncation, the total number of generated terms remains well below the theoretical limit for degree-5 monomials in 128 dimensions, indicating that the constructed generators remain sparse.

### 5.3. Pruning the generators

Figure 4 demonstrates that pruning a large fraction of polynomials can significantly degrade accuracy, especially in early layers. On the other hand, moderate pruning largely preserves performance. Figure 6 further confirms the validity of the approach outlined in Figure 2: pruning polynomials suffices to reduce the number of monomials significantly.

### 5.4. The parameter-efficiency of VI-Nets

We show that VI-Net achieves parameter efficiency by pruning up to a moderate number of polynomials. We create four variants by truncating increasing portions of the trailing layers, substituting them with the polynomial expansions: VI-Net–Tiny (retains roughly 30% of base layers), VI-Net–Small (roughly 50%), VI-Net–Medium (roughly 80%), and VI-Net–Large (removes only the last layer). Ta-

*Table 1.* Elapsed time, number of polynomials and monomials generated by the vanishing ideal algorithm vs. percentage of removed Layers: We compare the effect of removing different fractions of the final layers on the vanishing ideal algorithm.

| CIFAR-10 (ResNet-18) | | | |
|---|---|---|---|
| % of Layers Removed | Monomials | Polynomials | Time (s) |
| 64.5% | 8,386 | 53,132 | 6,468.36 |
| 53.3% | 4,462 | 14,297 | 245.09 |
| 46.7% | 1561 | 4,346 | 11.73 |
| 26.7% | 167 | 1,319 | 7.31 |
| 6.7% | 130 | 1,280 | 6.94 |
| CIFAR-100 (ResNet-34) | | | |
| % of Layers Removed | Monomials | Polynomials | Time (s) |
| 75.8% | 7,625 | 188,743 | 3368.76 |
| 50.1% | 5,196 | 53,675 | 82.46 |
| 43.9% | 1,990 | 30,004 | 18.38 |
| 25.1% | 196 | 12,978 | 7.46 |
| 7.1% | 237 | 12,997 | 8.56 |

ble 2 shows that VI-Nets often achieve both higher throughput and competitive performance.

*Table 2.* Performance Comparison for VI-Net. Total Parameters is the sum of Baseline Parameters (truncated ResNet) and Polynomial Parameters (generated expansions), measured in millions (M). Throughput is measured on the test split (batch size 256), averaged across 5 random seeds with standard deviation indicated.

| CIFAR-10 (ResNet-18) | | | | | |
|---|---|---|---|---|---|
| Model | Total | Base | Poly | Acc (%) | Throughput (img/s) |
| VI-Net18-Tiny | 1.86M | 0.68M | 1.18M | 88.62 | 100798 ± 20506 |
| VI-Net18-Small | 2.84M | 2.18M | 0.66M | 92.66 | 79307 ± 15576 |
| VI-Net18-Medium | 4.35M | 3.96M | 0.39M | 92.92 | 71851 ± 13987 |
| VI-Net18-Large | 9.20M | 8.81M | 0.39M | 93.02 | 62086 ± 11291 |
| ResNet18 | 11.24M | 11.24M | - | 92.89 | 66533 ± 12577 |
| CIFAR-100 (ResNet-34) | | | | | |
| Model | Total | Base | Poly | Acc (%) | Throughput (img/s) |
| VI-Net34-Tiny | 3.52M | 2.85M | 0.67M | 71.94 | 42064 ± 7247 |
| VI-Net34-Small | 5.88M | 5.21M | 0.67M | 74.03 | 37611 ± 6502 |
| VI-Net34-Medium | 14.60M | 14.20M | 0.40M | 74.66 | 29285 ± 5791 |
| VI-Net34-Large | 19.32M | 18.92M | 0.40M | 74.78 | 27862 ± 5358 |
| ResNet34 | 20.35M | 20.35M | - | 74.75 | 28253 ± 4290 |

## 6. Conclusion and Limitations

We establish a connection between manifold learning and computational algebra, demonstrating how vanishing ideal algorithms can be used to accurately discriminate different classes in the latent spaces of neural networks, despite said spaces being of much higher dimensionality than what is typically considered in the context of vanishing ideal algorithms. By proposing VI-Net, we showed that the found generators of the respective vanishing ideals can be used to construct a polynomial layer that, when appended to a the truncated smaller network, can match the performance of

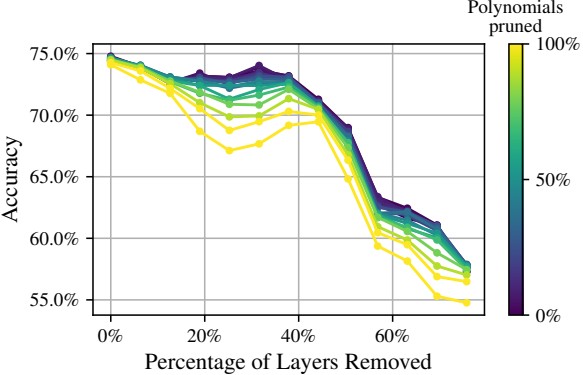

*Figure 4.* CIFAR-100 (ResNet-34): Test accuracy vs. percentage of removed layers and the pruning fraction of polynomials. We compare the performance of different pruning ratios for removing different fractions of layers.

the full network with higher parameter-efficiency.

However, limitations remain. If the latent-space class manifolds overlap significantly or have high intrinsic dimensionality, the polynomials may fail to capture them accurately. Exploring higher-dimensional latent spaces, such as those in transformer models, could be a promising direction for future work.

## Impact Statement

This paper presents work that aims to advance the field of Machine Learning by introducing scalable and efficient methods for analyzing latent spaces in neural networks using vanishing ideals. While this research primarily focuses on foundational advancements with potential applications in interpretability, efficiency, and generalization, we acknowledge that any progress in Machine Learning can have both positive and negative societal impacts. For this work, we do not identify specific ethical concerns or societal consequences that warrant further discussion. However, we encourage the community to consider the broader implications of deploying these methods in various domains.

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

# A. Appendix: Additional details, experiments and figures

### A.1. Implementation Details

For the parameter-efficient VI-Net implementation, we prune to a fixed number of polynomials and monomial terms. Each variant includes a PCA layer reducing to 128 components, encoded monomials ($512 \times 128$), polynomial coefficients ($1280 \times 512$ or $12800 \times 512$), and a final linear layer.

*Table 3.* Name of the layer at which we truncate the baseline NN.

| Variant | ResNet18 (CIFAR10) | ResNet34 (CIFAR100) |
|---------|--------------------|--------------------|
| **Tiny** | `layer2.1.bn2` | `layer3.1.bn1` |
| **Small** | `layer3.1.bn1` | `layer3.3.bn1` |
| **Medium** | `layer4.0.bn1` | `layer4.1.bn1` |
| **Large** | `layer4.1.bn1` | `layer4.2.bn1` |

All ResNet18-based variants use 512 monomials and 1280 polynomials, while ResNet34-based variants use 512 monomials and 12800 polynomials.

## A.2. Additional Experiments

We list additional experiments.

### A.2.1. IMPACT OF TRUNCATION ON CIFAR-10

We apply truncations to a standard ResNet-18, retaining 512 training images per class for constructing approximate vanishing ideals. We do not prune polynomials in this experiment. Both the linear layer and the coefficients were retrained for 20 epochs.

Figure 5 shows that removing up to 20% of the final convolutional layers does not degrade the performance of the linear head compared to VI-Net, likely because the data is already linearly separable at these layers. Additionally, we observe that a network with randomly generated monomials performs significantly worse than the linear head when removing between 0% and 30% of the final layers. However, as more layers are removed, the non-linearity introduced by the monomials becomes beneficial. Notably, VI-Net maintains higher accuracy and consistently outperforms both the linear head and the randomly sampled monomials.

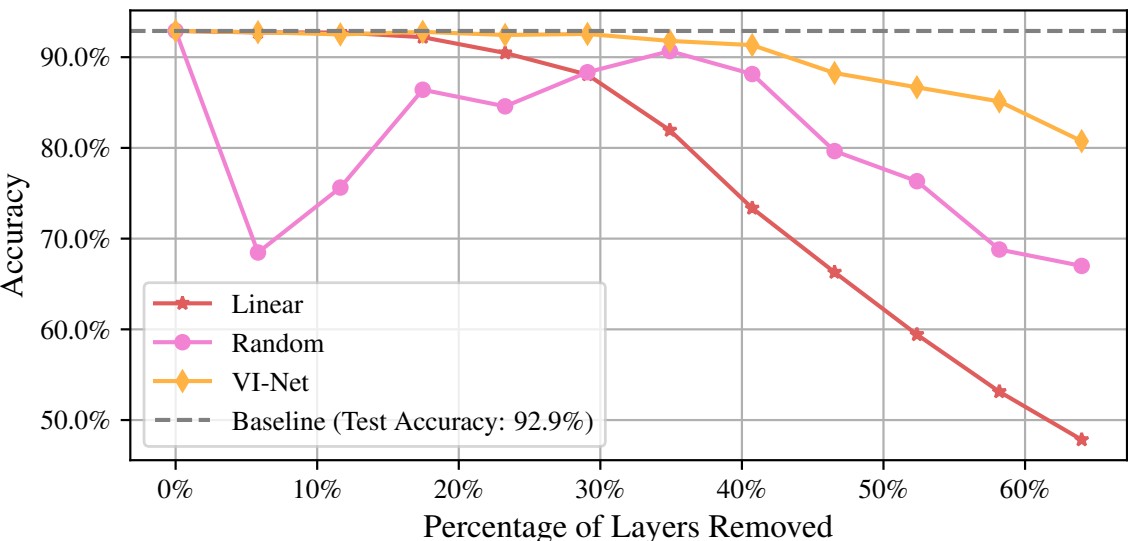

*Figure 5.* CIFAR-10 (ResNet-18): Test accuracy vs. percentage of removed convolutional layers. The dashed gray line indicates the fully trained baseline. A linear layer retains accuracy only with minimal truncation, whereas VI-Net maintains high accuracy even with substantial truncation.

### A.2.2. PRUNING AND MONOMIAL COUNT

In this experiment, we examine how the number of monomials required to evaluate the generators varies with different pruning ratios. Figure 6 shows that pruning polynomials significantly reduces the number of monomials to be evaluated. Notably, when removing between $20\%$ and $40\%$ of the layers, the number of remaining monomials falls below the latent dimension of 128 (after PCA). This explains the increased susceptibility of these layers to accuracy degradation: excessive pruning of polynomials leads to the loss of too many monomial terms, reducing accuracy. Furthermore, this result suggests that ABM efficiently constructs representations in this range, as it retains relatively few monomial terms without requiring explicit pruning.

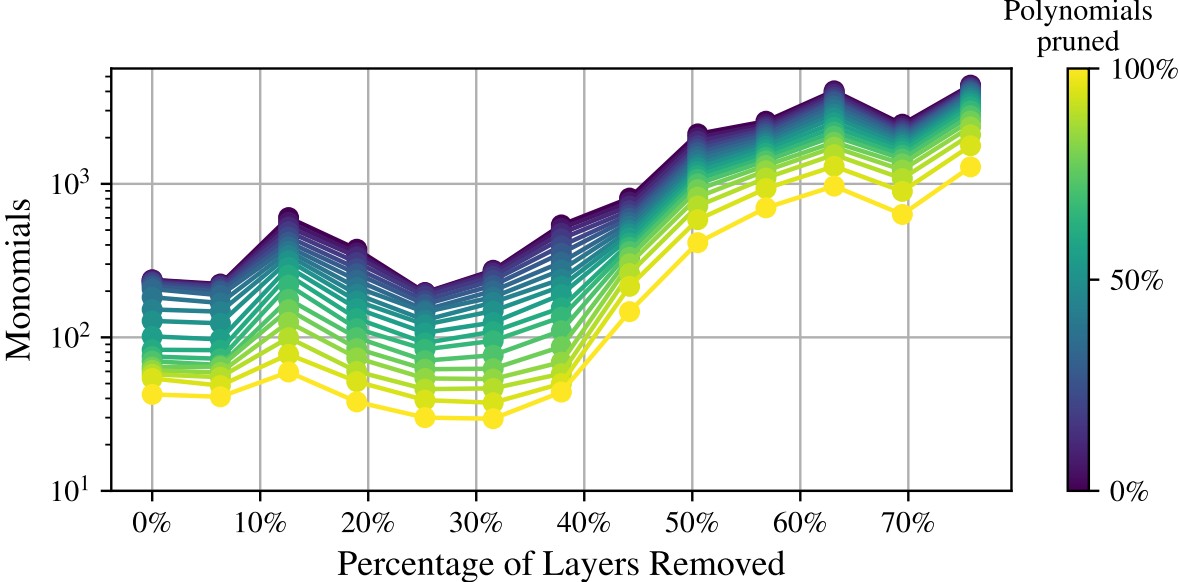

*Figure 6.* CIFAR-100 (ResNet-34): Impact of pruning on monomial count. Removing a large fraction of polynomials reduces the number of remaining monomials, sometimes below the latent dimension of 128. This reduction explains the observed accuracy drop at high pruning ratios, particularly when removing $20\%$ to $50\%$ of the layers (see Figure 4).

### A.2.3. COMPARISON OF DIFFERENT VANISHING IDEAL ALGORITHMS

We construct VI-Net following the pipeline in Algorithm 1, using different vanishing ideal algorithms. Specifically, we compare ABM to OAVI with two oracle variants: Frank-Wolfe and Accelerated Gradient Descent. The baseline network is a ResNet34, trained and tested on CIFAR-100.

Figure 7 shows that all vanishing ideal algorithms yield similar performance for the resulting VI-Nets, with ABM slightly outperforming the OAVI variants in intermediate layers. This similarity suggests that the algorithms capture comparable structural properties of the data. Additionally, since we retrain the coefficients of the constructed generators, the primary difference lies in the selection of monomial terms by each algorithm.

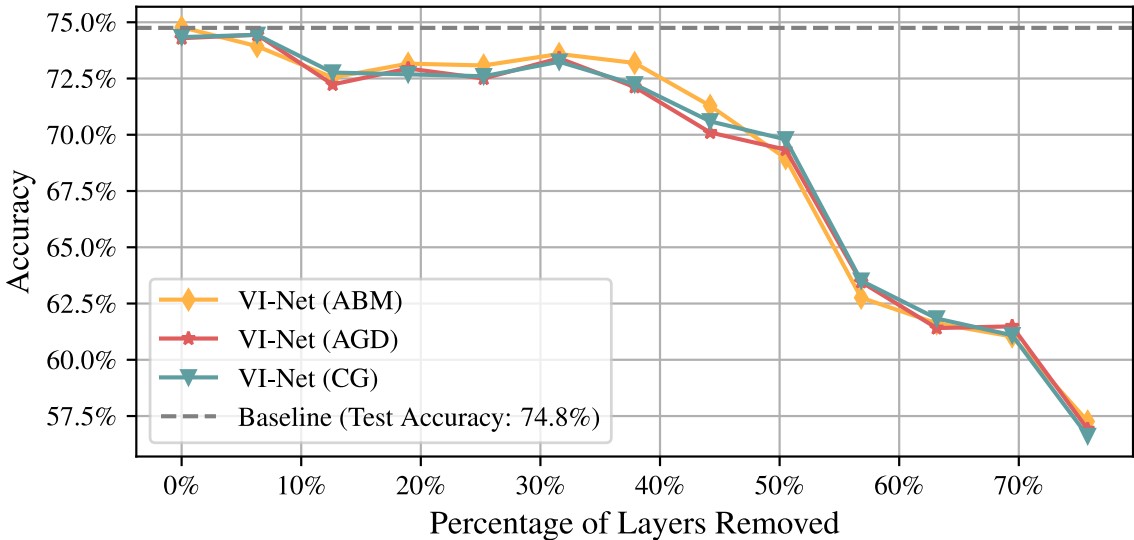

*Figure 7.* CIFAR-100 (ResNet-34): Test accuracy vs. percentage of removed convolutional layers. The dashed gray line indicates the fully trained baseline. We compare different VI-Nets, constructing polynomials using ABM, as well as OAVI with Accelerated Gradient Descent and Frank-Wolfe.

### A.2.4. Effect of Pruning on CIFAR-10 (ResNet18)

In this experiment, we examine the effect of polynomial pruning at different fractions on ResNet18 trained on CIFAR-10. Figure 8 shows that removing up to $40\%$ of all layers has a negligible impact on accuracy, suggesting that many polynomials are not essential for classification and can be discarded. A significant accuracy drop is observed only at extreme pruning levels ($> 90\%$). In earlier layers, pruning reduces performance more noticeably, likely due to the loss of expressivity, which the polynomial layer would otherwise compensate for.

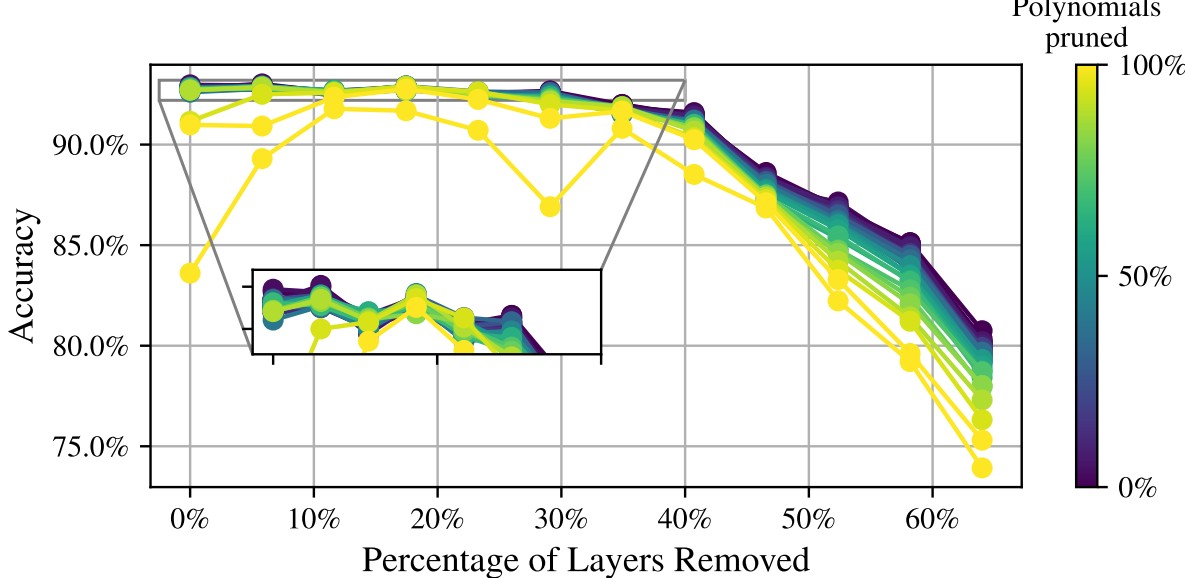

*Figure 8.* CIFAR-10 (ResNet18): Impact of polynomial pruning on test accuracy. Accuracy remains stable up to moderate pruning levels, but excessive pruning—especially in early layers—leads to a significant drop due to reduced expressivity.

A.2.5. EFFECT OF PRUNING ON OAVI-CG BASED VI-NET

We analyze the effect of pruning the generator set for OAVI using Conditional Gradients (Frank-Wolfe) as the oracle. The results exhibit a similar trend to Figure 4, though the accuracy shows a slightly weaker dependence on the pruning ratio. This is likely due to the higher number of constructed generators, which provides additional redundancy.

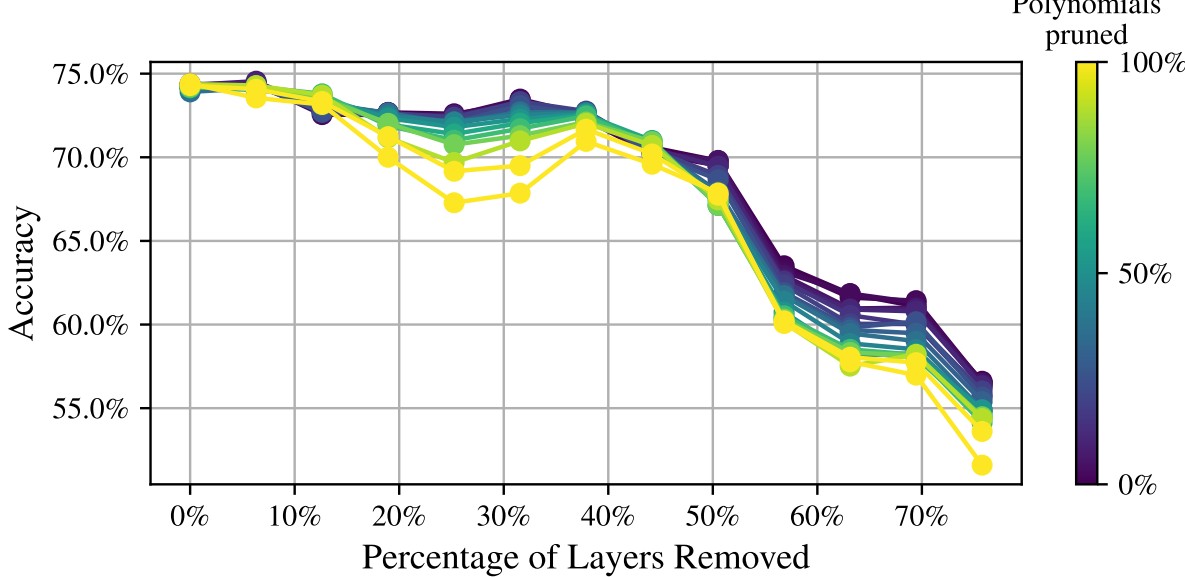

*Figure 9.* CIFAR-100 (ResNet34): Impact of polynomial pruning on test accuracy for OAVI-CG. While the general trend mirrors Figure 4, the higher number of generators reduces sensitivity to pruning.

A.2.6. EFFECT OF PRUNING ON OAVI-AGD BASED VI-NET

We analyze the effect of pruning the generator set for OAVI using Accelerated Gradient Descent as the oracle. The results exhibit a similar trend to Figure 4, though the accuracy shows a slightly weaker dependence on the pruning ratio. This is likely due to the higher number of constructed generators, which provides additional redundancy.

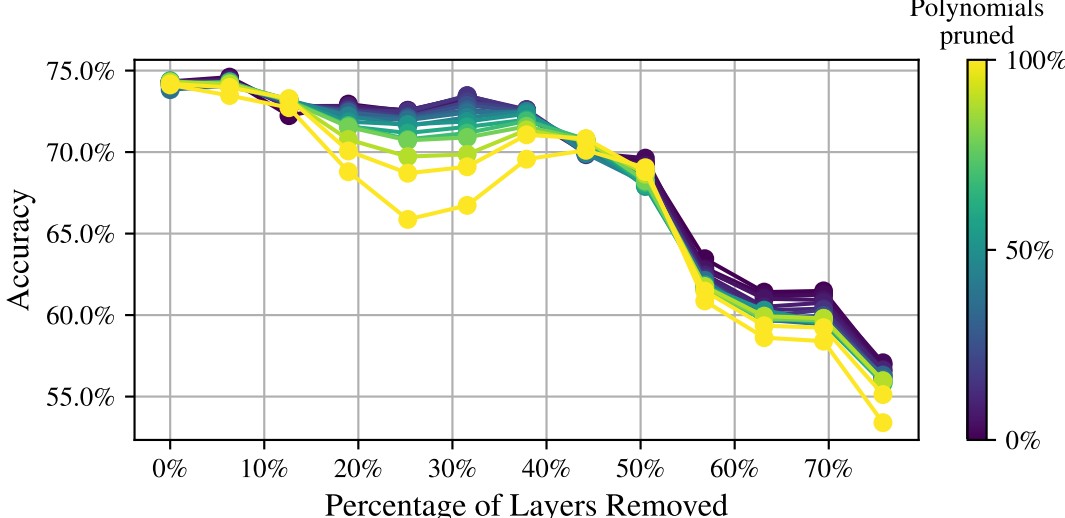

*Figure 10.* CIFAR-100 (ResNet-34): Impact of polynomial pruning on test accuracy for OAVI-AGD. The overall trend resembles Figure 4, though the higher number of generators reduces sensitivity to pruning.

A.2.7. TRUNCATION VS. NUMBER OF GENERATORS

We analyze the number of polynomials and monomials constructed by ABM when removing different layers of ResNet34 on CIFAR-100. A clear trend emerges: earlier layers produce more monomials and polynomials, increasing computational cost. However, at the start of residual blocks, we observe an unusually high number of constructed generators, suggesting a shift in the algebraic structure of the data at these layers.

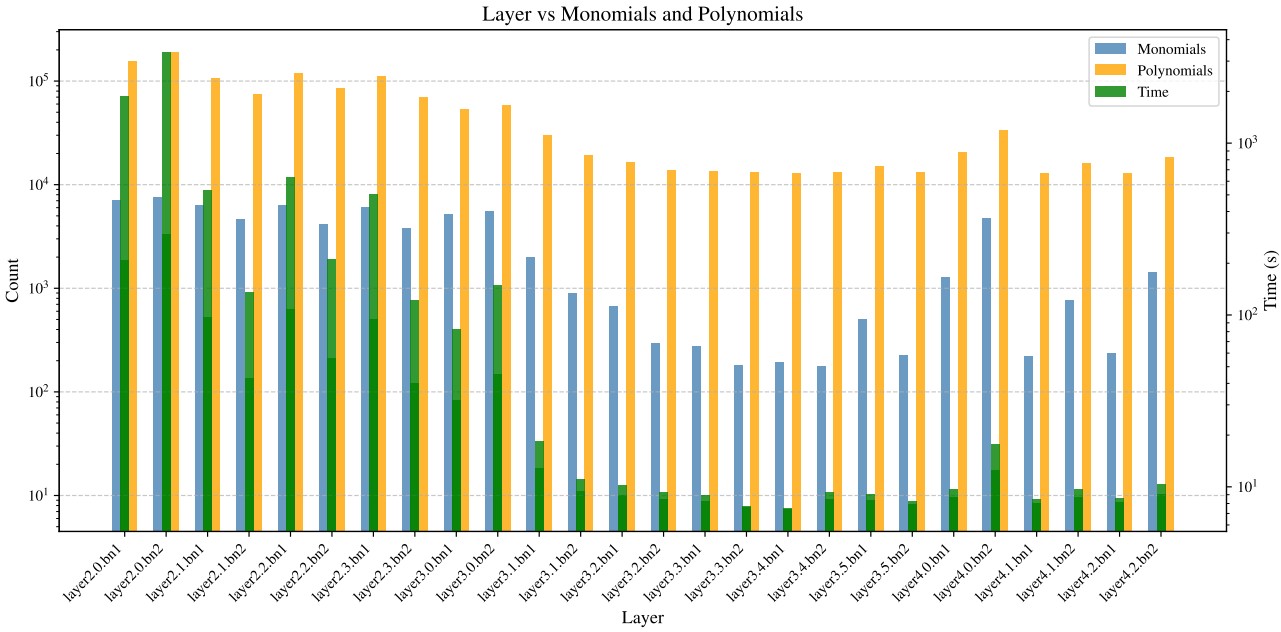

Figure 11. CIFAR-100 (ResNet34): Number of constructed monomials and polynomials across network layers. Earlier layers generate more terms, increasing computation time, while residual block boundaries show an unexpected increase in generators, indicating structural changes in the data.

A.2.8. Intrinsic Dimension vs. Vanishing Ideal

In this experiment, we analyze the intrinsic dimension of the data manifolds at different layers of ResNet34 and its effect on the number of constructed monomials. We estimate the intrinsic dimension using the Levina-Bickel estimator (Levina & Bickel, 2004) with the MacKay-Ghahramani extension (MacKay & Ghahramani, 2005).

Figure 12 suggests a strong correlation between intrinsic dimension and the number of constructed monomial terms. This aligns with intuition: a more complex underlying manifold requires a more intricate algebraic variety, necessitating a larger number of monomial terms.

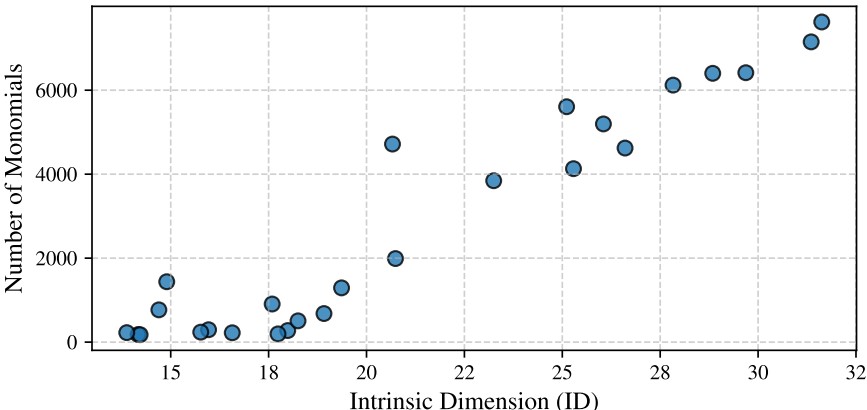

*Figure 12.* CIFAR-100 (ResNet34): Correlation between intrinsic dimension and number of monomial terms. Higher intrinsic dimensionality corresponds to a greater number of constructed monomials, reflecting increased algebraic complexity.

# B. Appendix: Theoretical Analysis and Learning Guarantees

## B.1. Proofs

We provide the proof for the theoretical results.

### B.1.1. InEx-ACTIVATION FUNCTION

The InEx activation function is a theoretical tool that we introduce to encode a VI-Net in the form of Definition 4.1. For illustration, we compute several monomials.

**Example B.1.** Consider the degree 2 monomial $xy$. This can be expressed as

$$xy = \frac{1}{2}\left((x+y)^2 - x^2 - y^2\right),$$

which requires only addition and squaring operations. We generalize this approach to degree 3 monomials using the inclusion-exclusion principle. For example, the monomial $xyz$ can be computed by considering all subsets of the variables $\{x, y, z\}$ and their contributions to the cube expansion:

$$(x+y+z)^3 = x^3 + y^3 + z^3 + 3x^2y + 3x^2z + 3y^2x + 3y^2z + 3z^2x + 3z^2y + 6xyz.$$

To isolate $xyz$, we subtract terms involving lower-order monomials:

$$xyz = \frac{1}{6}\left((x+y+z)^3 - (x+y)^3 - (x+z)^3 - (y+z)^3 + x^3 + y^3 + z^3\right).$$

This approach also applies to monomials where variables appear with higher powers, such as $x^2y$. To handle such cases, we replace one variable with a duplicate of another. For instance, setting $z = x$ allows us to rewrite $x^2y$ in terms of the cube expansion above.

The following lemmas formalize Example B.1.

**Lemma B.2** (Multiplication with InEx-activation). *Let $\alpha^d$ be given as in Definition 4.3. Let $x_1, \ldots, x_d \in \mathbb{R}$ and let $(I_1, \ldots, I_{2^d})$ be the tuple of all subsets of $[d]$, ordered by cardinality. Then, for $\mathbf{z}$ with $z_j = \sum_{i \in I_j} x_i$ it holds*

$$\alpha^d(\mathbf{z}) = \prod_{i=1}^{d} x_i.$$

*Proof.* This is an application of the inclusion-exclusion-principle. Note, that by the multinomial theorem, it holds

$$\left(\sum_{i \in I_j} x_i\right)^d = n! \sum_{\substack{\sum_{i \in I_j} k_i = n}} \left(\prod_{i \in I_j} \frac{x_i^{k_i}}{k_i!}\right).$$

Therefore a monomial of degree $d$ is in the support of $\left(\sum_{i \in I_j} x_i\right)^d$ if and only if all variable indices are in $I_j$. Further, the coefficient does not depend on the cardinality of $I_j$, which allows to apply the inclusion-exclusion principle to the sum. Thus, after rescaling by $\frac{1}{n!}$ only the term $\prod_{i=1}^{d} x_i$ survives. $\square$

**Lemma B.3** (Computing monomials with InEx-activation). *Let $x_1, \ldots, x_k \in \mathbb{R}$ and $\alpha_i \geq 1$ be given such that $\sum_{i=1}^{k} \alpha_i = d$. Then, there exists a matrix $\mathbf{W}$ with $k$ columns and $2^d$ rows, such that*

$$\alpha^d(\mathbf{W}\mathbf{x}) = \prod_{i=1}^{k} x_i^{\alpha_i}.$$

*Proof.* First introduce artificial variables $x_{i,1}, \ldots, x_{i,\alpha_i}$, which we all set to $x_i$. Define the set $I = \{(i, q) : i \in [k], q \in [\alpha_i]\}$, which has cardinality $d$, since $\sum_i^k \alpha_i = d$. Then, we can rewrite this product as $\prod_{i=1}^k x_i^{\alpha_i} = \prod_{(i,q) \in I} x_{i,q}$. Let $I_j \subseteq I$ be one of the $2^d$ subsets of $I$ and define $\mathbf{w}_j^T := \sum_{(i,q) \in I_j} e_i$, where $e_i$ denotes the $i$-th basic vector in $k$ dimensions. Since $x_{i,q} = x_i$ for all $q \leq \alpha_i$ it holds that $\mathbf{w}_j^T \mathbf{x} = \sum_{(i,q) \in I_j} x_{i,j}$. Therefore, if we order the sets $I_j$ by cardinality, we can apply Lemma B.2 to obtain for $\mathbf{W} = (\mathbf{w}_1, \ldots, \mathbf{w}_{2^d})^T$ that $\alpha^d(\mathbf{W}\mathbf{x}) = \prod_{(i,q) \in I} x_{i,q} = \prod_{i=1}^k x_i^{\alpha_i}$. $\qquad\square$

**Lemma B.4** (Lipschitz property of InEx-activation). *Let $\alpha^d$ be given as in Definition 4.3. Then, $\alpha^d$ is $d$-Lipschitz on the image of $\mathbf{W}$ for $\mathbf{x} \in [-1, 1]^k$ from Lemma B.3.*

*Proof.* From Lemma B.3, we know that the mapping $\mathbf{x} \to \alpha^d(\mathbf{W}(\mathbf{x}))$ is equivalent to the mapping of the corresponding monomial, which is $d$-Lipschitz on $[-1, 1]^k$. Next, for each $i \in [k]$, it holds $e_i^T$ is a row in $\mathbf{W}$, therefore for $\mathbf{x}, \mathbf{x}' \in [-1, 1]^k$ it holds $\|\mathbf{W}\mathbf{x} - \mathbf{W}\mathbf{x}'\|_2 \geq \|\mathbf{x} - \mathbf{x}'\|_2$. Since arbitrary points in the image of $\mathbf{W}$ can be expressed as $\mathbf{W}\mathbf{x}$ with $\mathbf{x} \in [-1, 1]^k$ we obtain

$$\frac{\|\alpha^d(\mathbf{W}\mathbf{x}) - \alpha^d(\mathbf{W}\mathbf{x}')\|_2}{\|\mathbf{W}\mathbf{x} - \mathbf{W}\mathbf{x}'\|_2} \leq \frac{\|\alpha^d(\mathbf{W}\mathbf{x}) - \alpha^d(\mathbf{W}\mathbf{x}')\|_2}{\|\mathbf{x} - \mathbf{x}'\|_2}$$
$$\leq \frac{d\|\mathbf{x} - \mathbf{x}'\|_2}{\|\mathbf{x} - \mathbf{x}'\|_2}$$
$$= d.$$

Here, the second inequality is due to the Lipschitz property of the map $\alpha^d(\mathbf{W}(\cdot))$. $\qquad\square$

### B.1.2. ENCODING A VI-NET AS DEFINITION 4.1

By the previous section, we know that we can compute monomials employing the InEx activation function and a matrix $\mathbf{W}_M$.

**Lemma B.5.** *Assume $G$ is given as in Assumption 4.2. Then, $G$ can be encoded as the two layer NN*

$$\mathbf{W}_C(\sigma(\mathbf{W}_M(\mathbf{x}))),$$

*where $\mathbf{W}_M \in \mathbb{R}^{s \times n}$, and we stack InEx-activation functions such that $\sigma = (\alpha^{d_1}, \ldots, \alpha^{d_S}) : \mathbb{R}^s \to \mathbb{R}^S$ and $\mathbf{W}_C \in \mathbb{R}^{N \times S}$ such that $s \in \mathcal{O}(S)$.*

*Proof.* Construct the matrix $\mathbf{W}_M$ by employing the matrices $\mathbf{W}$ from Lemma B.3 for each monomial to compute and stacking the respective InEx activation functions, such that $\sigma = (\alpha^{d_1}, \ldots, \alpha^{d_S})$. Thus, the $i$-th component of the vector $\sigma(\mathbf{W}_M\mathbf{x})$ will be the $i$-th monomial of the $S$ polynomials we have to compute to evaluate $G$. Finally, we construct $\mathbf{W}_C$ by choosing the rows as the coefficient vectors of the polynomials in $G$. This concludes the proof. $\qquad\square$

### B.1.3. SPECTRAL PROPERTIES OF VI-NET ENCODING

In this section, we obtain bounds for the norm of the matrices that are used to encode a VI-Net in the format of Definition 4.1. We begin my analysing the properties of $\mathbf{W}_M$

**Lemma B.6** (Spectral properties of $\mathbf{W}_M$). *Let Assumption 4.2 hold, and let $\mathbf{W}_M$ be constructed as in Lemma B.2. Then, it holds that:*

$$\|\mathbf{W}_M\|_2 \leq 2^{\frac{d}{2}} d\sqrt{S} \quad and \quad \frac{\|\mathbf{W}_M^T\|_{2,1}}{\|\mathbf{W}_M\|_2} \leq 2^d S,$$

*where $S$ is the number of monomials, $n$ is the number of variables and $d$ the highest degree of any one monomial.*

*Proof.* We bound the Frobenius norm of the matrix $\mathbf{W}_M$. By construction, $\mathbf{W}_M$ has at most $S \cdot 2^d$ rows. For each row, the squared euclidean norm is at most $d^2$. Therefore, it holds

$$\|\mathbf{W}_M\|_F \leq \sqrt{S2^d \cdot d^2} = 2^{\frac{d}{2}} d\sqrt{S}. \tag{1}$$

As $\|\mathbf{W}_M\|_2 \leq \|\mathbf{W}_M\|_F$, it follows that

$$\|\mathbf{W}_M\|_2 \leq 2^{\frac{d}{2}} d\sqrt{S}. \tag{2}$$

Next, for the $\ell_{2,1}$ norm, recall that

$$\|\mathbf{W}_M^T\|_{2,1} = \sum_{i=1}^m \|\mathbf{W}_{M,i}^T\|_2, \tag{3}$$

where $\mathbf{W}_{M,i}^T$ is the $i$-th column of $\mathbf{W}_M^T$, i.e., the $i$-th row of $\mathbf{W}_M$. Let $\mathbf{w}$ denote the row with the largest euclidean norm in $\mathbf{W}_M$. This row is part of a block required for computing a specific degree $d$ monomial $\prod_{i=1}^k x_i^{\alpha_i}$ as in Lemma B.3. Let $j$ denote the index with the largest entry in $\mathbf{w}$, i.e., $j \in \operatorname{argmax}_{i \in [k]} \alpha_i$. For each of the other variable $i'$ there is a row of the form $\alpha_{i'} e_{i'} + \alpha_j e_j$. Thus, there are $k - 1$ such rows, i.e, in the $j$-th column of $\mathbf{W}$, the term $\alpha_j$ occurs $k - 1$ times, one for each of the remaining variables. By our choice of $j$ it holds

$$\sum_{i=1}^k \alpha_i^2 \leq (k-1)\alpha_j^2 + \alpha_j^2 \leq \|\mathbf{W}_{M,j}\|_2^2.$$

Therefore, it holds that the largest euclidean norm of any row is bounded by the largest euclidean norm of any column. Note that the spectral norm of a matrix is bounded from below by the column with the largest euclidean norm,i.e.,

$$\|\mathbf{W}_M\|_2 \geq \max_j \|\mathbf{W}_{M,:,j}\|_2. \tag{4}$$

Thus, we conclude that $\|\mathbf{W}_M\|_{2,1} \leq 2^d S \|\mathbf{W}_M\|_2$ and therefore $\frac{\|\mathbf{W}_M\|_{2,1}}{\|\mathbf{W}_M\|_2} \leq 2^d S$. □

**Lemma B.7** (Norm properties of $\mathbf{W}_C$). *Let $\mathbf{W}_C \in \mathbb{R}^{N \times S}$ have rows bounded in the $\ell_1$ norm by $\tau$. Then, it holds*

$$\|\mathbf{W}_C\|_2 \leq \sqrt{N}\tau \quad and \quad \frac{\|\mathbf{W}_C^T\|_{2,1}}{\|\mathbf{W}_C\|_2} \leq N\sqrt{S},$$

*where $S$ is the number of monomials and $N$ is the number of polynmials.*

*Proof.* We have

$$\|\mathbf{W}_C\|_2 \leq \sqrt{N}\|\mathbf{W}_C\|_\infty \tag{5}$$
$$\leq \sqrt{N}\tau. \tag{6}$$

For the ratio, it holds

$$\frac{1}{\sqrt{S}}\|\mathbf{W}_C\|_\infty \leq \|\mathbf{W}_C\|_2.$$

For any vector, its euclidean norm is always bounded by the $\ell_1$-norm. Therefore, it holds

$$N\|\mathbf{W}_C\|_\infty \geq \|\mathbf{W}_C^T\|_{2,1}.$$

Combining these results, we obtain

$$\frac{\|\mathbf{W}_C^T\|_{2,1}}{\|\mathbf{W}_C\|_2} \leq \frac{N\|\mathbf{W}_C\|_\infty}{\frac{1}{\sqrt{S}}\|\mathbf{W}_C\|_\infty}$$
$$= N\sqrt{S}$$

□

B.1.4. PROOF OF THEOREM 4.5

Finally, we are ready to prove Theorem 4.5.

**Theorem B.8.** *Let $\phi$ be an NN with $L$ layers in the form Definition 4.1. Let $G$ be a polynomial layer as in (POLY) satisfying Assumption 4.2, and $\tilde{\phi}$ the corresponding VI-Net based on the truncated NN $\hat{\phi}$ with $L'$ layers. Then, the spectral complexity of $\tilde{\phi}$ can be bounded as follows*

$$
R_{\tilde{\phi}} = \frac{d \prod\limits_{i \in \{M,C,F\}} \|\mathbf{W}_i\|_2}{\prod\limits_{i=L'+1}^{L} \rho_i \|\mathbf{W}_i\|_2} \left( \frac{\sum\limits_{\substack{i=1 \\ i \in \{M,C,F\}}}^{L'} \frac{\|\mathbf{W}_i^T\|_{2,1}^{\frac{2}{3}}}{\|\mathbf{W}_i\|_2^{\frac{2}{3}}}}{\sum\limits_{i=1}^{L} \frac{\|\mathbf{W}_i^T\|_{2,1}^{\frac{2}{3}}}{\|\mathbf{W}_i\|_2^{\frac{2}{3}}}} \right)^{\frac{3}{2}} R_{\phi}.
$$

*Further, assume the linear layer satisfies $\|\mathbf{W}_F\|_2 \leq \lambda_1$ and $\frac{\|\mathbf{W}_F\|_{2,1}}{\|\mathbf{W}_F\|_2} \leq \lambda_2$. Then, it holds*

$$
\prod_{i \in \{M,C,F\}} \|\mathbf{W}_i\|_2 \leq 2^d d\tau \lambda_1 \sqrt{NS},
$$

$$
\sum_{i \in \{M,C,F\}} \frac{\|\mathbf{W}_i^T\|_{2,1}^{\frac{2}{3}}}{\|\mathbf{W}_i\|_2^{\frac{2}{3}}} \leq 2^{\frac{2d}{3}} S^{\frac{2}{3}} + N^{\frac{2}{3}} S^{\frac{1}{3}} + \lambda_2^{\frac{2}{3}}.
$$

*Proof.* We employ the absolute value function in $G$ as an activation function, which is 1-Lipschitz and the InEx activation function, which is $d$-Lipschitz on its domain, while the last linear layer has no activation function. Using our encoding from Lemma 4.4, we get that the spectral complexity of this encoding is given by

$$
R_{\tilde{\phi}} = \left( d \prod_{i \in \{M,C,F\}} \|\mathbf{W}_i\|_2 \right) \left( \prod_{i=1}^{L'} \rho_i \|\mathbf{W}_i\|_2 \right)
$$
$$
\cdot \left( \sum_{\substack{i \in \{1,\ldots,L'\} \\ i \in \{M,C,F\}}} \frac{\|\mathbf{W}_i^T\|_{2,1}^{2/3}}{\|\mathbf{W}_i\|_2^{2/3}} \right)^{3/2} .
$$

The spectral complexity of the baseline NN is given by the definition as

$$
R_{\phi} = \left( \prod_{i=1}^{L} \rho_i \|\mathbf{W}_i\|_2 \right) \left( \sum_{i=1}^{L} \frac{\|\mathbf{W}_i^T\|_{2,1}^{2/3}}{\|\mathbf{W}_i\|_2^{2/3}} \right)^{3/2} .
$$

Therefore it holds

$$
\frac{R_{\tilde{\phi}}}{R_{\phi}} = \frac{d \prod\limits_{i \in \{M,C,F\}} \|\mathbf{W}_i\|_2}{\prod\limits_{i=L'+1}^{L} \rho_i \|\mathbf{W}_i\|_2} \left( \frac{\sum\limits_{\substack{i=1 \\ i \in \{M,C,F\}}}^{L'} \frac{\|\mathbf{W}_i^T\|_{2,1}^{\frac{2}{3}}}{\|\mathbf{W}_i\|_2^{\frac{2}{3}}}}{\sum\limits_{i=1}^{L} \frac{\|\mathbf{W}_i^T\|_{2,1}^{\frac{2}{3}}}{\|\mathbf{W}_i\|_2^{\frac{2}{3}}}} \right)^{\frac{3}{2}} .
$$

Multiplication by $R_{\phi}$ yields the first result.
For the second part, we apply Lemma B.6 and Lemma B.7 and obtain

$$
\prod_{i \in \{M,C,F\}} \|\mathbf{W}_i\|_2 \leq 2^{\frac{d}{2}} d\sqrt{S} \cdot \sqrt{N}\tau \cdot \lambda_1.
$$

Reordering yields the result. For the sum, we proceed the same way. $\square$

In general, there is no a priori bound on the norms of the weight matrices, which is why we focussed on this comparison with the baseline network. We note that prior work (Pokutta et al., 2020; Zimmer et al., 2025) examined ways to train NN with bounded norms.

## C. Vanishing Ideal Algorithms

In this section, we provide an overview of the vanishing ideal algorithms considered in our setting: ABM and OAVI. Both algorithms take a set of points and construct generators of the (approximate) vanishing ideal. They extend noise-free generator construction algorithms, which require exact vanishing, by allowing polynomials to vanish up to a tolerance $\psi$. Structurally, they follow the same principles as exact algorithms but introduce robustness to noisy data.

Beyond returning a set of generators, denoted as $\mathcal{G}$ in this section, these algorithms also provide a set $\mathcal{O}$, known as the *order ideal*. An order ideal is a subset of monomial terms that is closed under division; that is, if a monomial such as $x^2y$ belongs to $\mathcal{O}$, then all its divisors (e.g., $1, x, y, x^2$) must also be in $\mathcal{O}$. A fundamental concept related to order ideals is the *border*, which captures monomials that are one step outside the order ideal.

**Definition C.1** (Border). Let $\mathcal{O} \subseteq \mathcal{T}$ be an order ideal. Then, the *d-border* of $\mathcal{O}$ is defined as:

$$\partial_d \mathcal{O} := \{u \in \mathcal{T}_d \mid t \in \mathcal{O}_{\leq d}, t \neq u, \text{ and } t \mid u\}.$$

A fundamental result from computational algebra states that every zero-dimensional ideal (such as the vanishing ideal of a finite set of points) admits a generating set known as a *border basis*. A border basis provides a structured way to represent the ideal while ensuring computational efficiency in polynomial interpolation and approximation.

**Definition C.2** (Border Basis). Let $\mathcal{G} = \{g_1, \ldots, g_m\}$ be a set of generators for a zero-dimensional ideal $I$, and let $\mathcal{O}$ be an order ideal. Then, $\mathcal{G}$ is a *border basis* of $I$ if:

1. $\mathcal{G}$ consists of polynomials of the form

$$g_u = u - \sum_{t \in \mathcal{O}} c_{u,t} t, \quad u \in \partial \mathcal{O},$$

where each $g_u$ is a polynomial whose leading term belongs to the border $\partial \mathcal{O}$.

2. The polynomials in $\mathcal{G}$ form a basis for the ideal $I$.

Intuitively, a border basis expresses each polynomial in the vanishing ideal in terms of monomials from the order ideal, ensuring that computations remain well-structured and stable. For a more detailed overview and the algebraic background we refer to Kehrein & Kreuzer (2006).

For this section, note that we think of sample of points $\mathbf{X}$ as a tuple, rather than a set, as we require that the order of points is fixed. Then, for a given monomial $u \in \mathcal{T}$ we write the *evaluation vector* $(u(\mathbf{x}_1), \ldots, u(\mathbf{x}_m))$ as $u(\mathbf{X})$. For a set of monomials, such as $\mathcal{O}$, we write the evaluation matrix $\mathcal{O}(\mathbf{X})$, where the entry indexed by $i$ and $j$ corresponds to the $i$-th monomial in $\mathcal{O}$ evaluated at the $j$-th point.

### C.1. Approximate Buchberger Möller Algorithm

The Approximate Buchberger-Möller Algorithm (ABM) constructs an approximate border basis, as described in (Limbeck, 2014). Given a term $u_i$ in the border, ABM computes the smallest eigenvalue and corresponding eigenvector of the matrix $A$, which contains the evaluations of all monomials at every point in the dataset. This process identifies the best-fitting polynomial relations while ensuring numerical stability. For a detailed explanation, see Limbeck (2014).

**Algorithm 2** Approximate Buchberger Möller Algorithm (ABM) (Limbeck, 2014)

**Input:** Dataset $\mathbf{X}$, a vanishing parameter $\psi \geq 0$
**Output:** An approximate $\mathcal{O}$-border basis

1: $\mathcal{O} := \{1\}, \mathcal{G} := \emptyset, d := 1, M := (1, \ldots, 1)^{\top} \in \mathbb{R}^{s,1}$
2: **while** $\partial_d \mathcal{O} = \{u_1, \ldots, u_l\} \neq \emptyset$ **do**
3:    **for** $i = 1$ to $l$ **do**
4:       $A := (u_i(\mathbf{X}), M)$
5:       $B := A^T A$
6:       $\lambda :=$ smallest eigenvalue of $B$
7:       **if** $\sqrt{\lambda} \geq \epsilon$ **then**
8:          $m := |\mathcal{O}|$
9:          $\mathbf{s} := (s_{m+1}, s_m, \ldots, s_1) :=$ the norm-one eigenvector of $B$ w.r.t. $\lambda$
10:          // Assume that $\mathcal{O} = [o_m, \ldots, o_1]$
11:          $g := s_{m+1} t_i + s_m o_m + \cdots + s_1 o_1$
12:          $\mathcal{G} \leftarrow \mathcal{G} \cup \{g\}$
13:       **else**
14:          $\mathcal{O} \leftarrow \mathcal{O} \cup \{u_i\}$
15:          $M := A$
16:       **end if**
17:    **end for**
18:    $d := d + 1$
19: **end while**
20: **return** $(\mathcal{G}, \mathcal{O})$

## C.2. Oracle Approximate Vanishing Ideal Algorithm

The Oracle Approximate Vanishing Ideal (OAVI) algorithm, introduced by Wirth & Pokutta (2022), incorporates a constrained convex optimization approach. Instead of computing the eigenvector corresponding to the smallest eigenvalue of $A^T A$, OAVI solves a constrained convex optimization problem at line 5 to determine the coefficient vector $\mathbf{c}$. This requires a convex optimization oracle and a constraint region, chosen as the $\ell_1$-ball of radius $\tau$. Like ABM, OAVI maintains a record of the evaluated monomials in the order ideal $\mathcal{O}$ within the matrix $A$. In Wirth & Pokutta (2022) the oracle was (among others) choosen as Conditional Gradient, which is known for constructing sparse solutions.

**Algorithm 3** Oracle Approximate Vanishing Ideal algorithm (OAVI) (Wirth & Pokutta, 2022)

**Input:** Dataset $\mathbf{X}$, vanishing parameter $\psi \geq 0$, coefficient bound $\tau \geq 2$
**Output:** The set of generators $\mathcal{G} \subseteq \mathcal{P}$ and the set $\mathcal{O} \subseteq \mathcal{T}$

1: $\mathcal{O} := \{1\}, \mathcal{G} := \emptyset, d := 1$
2: **while** $\partial_d \mathcal{O} = \{u_1, \ldots, u_l\} \neq \emptyset$ **do**
3:    **for** $i = 1, \ldots, l$ **do**
4:       $\ell \leftarrow |\mathcal{O}|, A \leftarrow \mathcal{O}(\mathbf{X}), \mathbf{b} = u_i(\mathbf{X}), P = \{\mathbf{y} \in \mathbb{R}^\ell : \|\mathbf{y}\|_1 \leq \tau - 1\}$
5:       $\mathbf{c} \in \text{argmin}_{\mathbf{y} \in P} \frac{1}{m} \|\mathbf{A}\mathbf{y} + \mathbf{b}\|_2^2$
6:       $g \leftarrow u_i + \sum_{j=1}^{\ell} c_j t_j$
7:       **if** $\text{MSE}(g, \mathbf{X}) \leq \psi$ **then**
8:          $\mathcal{G} \leftarrow \mathcal{G} \cup \{g\}$
9:       **else**
10:          $\mathcal{O} \leftarrow \mathcal{O} \cup \{u_i\}$
11:       **end if**
12:    **end for**
13:    $d \leftarrow d + 1$
14: **end while**

### C.3. Stochastic Versions of ABM and OAVI

In our experiments, a key bottleneck of the vanishing ideal algorithms was *memory*, primarily due to the matrix $A$ used in both methods. Given a dataset $\mathbf{X}$ with $m$ samples, the matrix $A$ has dimensions $m \times |\mathcal{O}|$, making storage and computation costly for large $m$.

To mitigate this, we employ a stochastic approach by subsampling the dataset, using only $m' \ll m$ samples to construct $A$. This significantly reduces memory usage while maintaining approximation quality. Additionally, we leverage lower precision arithmetic (e.g., float16) to further reduce the memory footprint without compromising stability.

### C.4. Addressing Data-Sensitivity of Vanishing Ideal Algorithms

Vanishing ideal algorithms are highly sensitive to the scale of the data. Recall that $\psi > 0$ specifies the extent to which polynomials are required to vanish on the training data (cf. Definition 2.3). This parameter depends on the scale of the data. Further, vanishing ideal algorithms typically require the data to be contained in $[-1, 1]^n$. Thus, we have to ensure that both the training data and, during inference, the input of the polynomial layer is properly scaled. To that end, we propose to employ a variant of tanh-rescaling (Hampel et al., 2011). Given the training dataset, we collect $\mu$ and $\sigma$ as the mean and standard deviation of the hidden activations at truncation, respectively. These can be easily calculated just by using forward passes and are collected similarly to the running averages in batch normalization. Then, a tanh-rescaling layer is applied right after truncation, mapping $z$ as

$$z \mapsto \tanh\left(\frac{z - \mu}{\sigma}\right).$$

This mapping ensures that the data is scaled to a reasonable range.

