# OpenReview forum: "Approximating Latent Manifolds in Neural Networks via Vanishing Ideals"
_ICML.cc/2025/Conference — ICML 2025 poster_

### Official Review · Reviewer_T8ev · 2025-02-25

**Overall Recommendation:** 3

**Summary:**

The authors propose neural networks motivated by the vanishing ideals, called VI-nets. They also investigate to obtain parameter-efficient representations of the vanishing ideal generators. They investigate the performance of the VI-nets theoretically and empirically.

**Claims And Evidence:**

The prposed method is based on the observation that low-dimensional manifolds that underlies data is characterized by vanishing ideals. They construct a neural network with polynomial activation functions based on this observation. The methodology seems reasonable.

**Essential References Not Discussed:**

N / A

**Experimental Designs Or Analyses:**

Although the authors empirically investigate the accuracy of the proposed method, I think investigating how the proposed method captures the underlying manifold for each class $k$ would be interesting. Is it possible to visualize or quantify how the learned polynomials separate the classes of the input data?

**Methods And Evaluation Criteria:**

In pratical situations, how to set the hyperparameters such as $L'$ and $S$ is not clear for me. Do you have any ideas about this?

**Other Comments Or Suggestions:**

N / A

**Other Strengths And Weaknesses:**

The authors encode the structure of data manifolds into neural networks using activation functions based on polynomials. The idea is interesting and relevent to the community.

**Questions For Authors:**

Please see the questions and concerns in Methods And Evaluation Criteria, Theoretical Claims, and Experimental Designs Or Analyses.

**Relation To Broader Scientific Literature:**

N / A

**Theoretical Claims:**

In section 4.2, the authors investigate the theoretical analysis of generalization error based on the existing results regarding spectral complexity. However, how the geometric aspect of the proposed method helps alleviating the generalization error is not clear for me. Indeed, the bound of $\kappa$ dependes on $2^d$, which becomes large if $d$ is large.

---

> ### Author Rebuttal · Authors · 2025-03-28
>
> Thank you for your review. We appreciate your feedback and will address your comments in detail.
>
> > In pratical situations, how to set the hyperparameters such as L' and S is not clear for me. Do you have any ideas about this?
>
> Thank you for your question. Our paper addresses this from two perspectives, one theoretical and the other one practical. From the theoretical perspective, we can compute for each $L'$ with Theorem 4.5 a "budget" in the form of a maximal number of monomials $S$, maximal degree $d$ and the other hyperparameters of our pipeline. This yields a provably lower spectral complexity and thus a better bound. From the practical perspective, our experiments indicate that we can remove about 40-50% of all convolutional layers before we observe a significant drop in performance. With the choice of $S$ you raise another interesting question, which we already addressed in the appendix (cf. Figure 6). There we analyse the monomial count ($S$). We found that it is not advisable to choose $S$ to be significantly lower than the latent dimensionality, otherwise we observe a large performance drop. We hope that this answers your question - if not, please let us know.
>
>
> > In section 4.2, the authors investigate the theoretical analysis of generalization error based on the existing results regarding spectral complexity. However, how the geometric aspect of the proposed method helps alleviating the generalization error is not clear for me. Indeed, the bound of $\kappa$ dependes on $2^d$, which becomes large if $d$ is large.
>
> Yes, you are right, the bound of $\kappa$ can become very loose if $d$ is large. However, in practice, we have always found it entirely sufficient to use values of $d$ not larger than $5$. To be more precise, we enforce the bound of $d$ as follows: we let the OAVI algorithm run until it terminates, and then we only take terms with maximum degree $d$. Despite setting $d$ to $5$, we never actually removed any terms from the ideal. In our computational setting, enforcing $d=5$ does in consequence not lead to any performance loss.
>
> > Although the authors empirically investigate the accuracy of the proposed method, I think investigating how the proposed method captures the underlying manifold for each class would be interesting. Is it possible to visualize or quantify how the learned polynomials separate the classes of the input data?
>
> We agree that it would be very interesting to investigate how the learned polynomials capture the underlying manifold. Since visualizations in this high-dimensional setting are generally not feasible, we provide the following alternative to see how two classes in particular are being separated. For this, one can choose for each of the class a vanishing polynomial according to our scoring function, say $p_1$ and $p_2$. Then, a simple way to visualize how $p_1$ and $p_2$ separate classes is by using a scatter plot with $x$-axis the absolute value of $p_1$ and the $y$-axis of $p_2$. We provide such a scatter plot here, showing how a training batch is becoming linearly separable: [https://imgur.com/o3VlqZl](https://imgur.com/o3VlqZl)
>
> As for a quantification of separability, this was the reason we introduced our scoring function. It essentially measures separability in the following sense: If the score is of the same order as the vanishing hyperparameter $\psi$, then a polynomial is not only vanishing on its own class but also on other classes. If the score is higher then $\psi$ then this indicates that the polynomial is not as approximately vanishing on other classes. Therefore the score introduced in line 195-197 in the second column is a direct way to quantify how well the learned polynomials separate the data.
>
>
>
> Thank you again for your review, we hope to have addressed your concerns. If you have any further questions, please let us know.

---

> > ### Comment · Reviewer_T8ev · 2025-04-03
> >
> > Thank you for your respense. I would like to keep my score.

---

### Official Review · Reviewer_9he7 · 2025-03-06

**Overall Recommendation:** 3

**Summary:**

The paper aims to improve parameter efficiency of neural networks by truncating a pretrained neural network, finding the polynomial generators of the vanishing ideal of (a finite sample of) the latent manifold, and transforming the latent space into a linearly separable feature space. The proposed method improves upon existing vanishing ideal algorithms, e.g. OAVI & ABM, by (1) performing PCA on latent space for dimensionality reduction, (2) pruning the found generators via class distinctiveness, and (3) other measures to improve computational efficiency such as stochastic sampling and lower-precision arithmetic. The paper also provides theoretical guarantee that the resulting VI-Net has lower spectral complexity and thus better generalization ability than the untruncated base model. The experiments on the CIFAR dataset demonstrate that VI-Net has comparable prediction accuracy to ResNet baselines while being more efficient.

**Claims And Evidence:**

The main claim of the paper is that, by replacing the final layers of a pretrained neural network with generators of vanishing ideals, VI-Net can achieve comparable prediction performance with fewer parameters. This is clearly supported by experimental evidence, particularly those from Table 2. It should be noted, however, that the improvement in computational efficiency is moderate in order to maintain the prediction accuracy (e.g. ~1.3x throughput on CIFAR-100).

**Essential References Not Discussed:**

I am not quite familiar with the literature in vanishing ideal algorithms. I'd say that the paper has provided sufficient references throughout the paper for readers to understand the context, but I'm not sure if the paper missed any related works that should have been discussed. Also, a dedicated section to related works would be helpful.

**Experimental Designs Or Analyses:**

While only restricted to one dataset, the experiments are comprehensive and demonstrate the effectiveness of different components of the proposed method. The effects of various factors such as truncation proportion, pruning ratio, etc, are investigated through various controlled experiments.

**Methods And Evaluation Criteria:**

The proposed method builds upon existing vanishing ideal algorithms such as OAVI and ABM. To improve the computational tractability of these algorithms, the authors first perform a PCA to reduce the latent space dimensionality. Then, the generators found by these algorithms are pruned based on how distinctive they are across samples from different classes. The monomials with zero coefficients in all remaining polynomial generators are also pruned to reduce the number of function evaluations. Overall, the proposed method is straightforward and it makes sense for the purpose of this paper -- to improve parameter efficiency of classification networks.

The evaluation criteria include the classification accuracy and computational efficiency (in terms of parameter count and inference throughput), which make sense for the proposed method. However, the evaluation is performed only on the CIFAR dataset. It would be better if the authors could also include other commonly used benchmarks for image classification.

**Other Comments Or Suggestions:**

Typos & potential confusions:
* L201: class $i$ --> class $j$
* L202: the $p$ in top $p\%$ could be confused with generator $p$
* L304: $\begin{pmatrix} i \\\\ d \end{pmatrix}$ should be the other way around?
* L323: What does $s \in \mathcal O(S)$ mean?
* L285: $\tilde \phi_{L'} \equiv \mathbf W_F\circ \mathbf W_C\ \circ\ ...$
* L392: Does elapsed time in Table 1 refer to the running time of the vanishing ideal algorithm?

**Other Strengths And Weaknesses:**

The paper is well-written. Important concepts are explained clearly and the flow is smooth and coherent.

**Questions For Authors:**

* Can we apply the methodology of VI-Net to regression tasks?
* Moreover, apart from building new models for prediction as you have done in this paper, can we use vanishing ideal algorithms to analyze the structure of the latent manifold itself and provide some kind of interpretability?

**Relation To Broader Scientific Literature:**

The paper mainly applies existing vanishing ideal algorithms, such as OAVI and ABS, to the NN latent space. It introduces some pre-/post-processing techniques to improve the computational tractability of these algorithms on high-dimensional spaces, but does not modify the vanishing ideal algorithms themselves.

**Theoretical Claims:**

Section 4 of the paper claims that VI-Net can achieve lower spectral complexity and thus better generalization ability than untruncated networks. Theorem 4.5 and Corollary 4.6 provide explicit formulas for the spectral complexity and the generalization error bound of VI-Net. The proof is provided in the Appendix and it looks correct to me. The only issue is that there seem to be no direct comparison between VI-Net and the base model. For example, under what condition does $\kappa < 1$, i.e. VI-Net has a lower spectral complexity?

---

> ### Author Rebuttal · Authors · 2025-03-31
>
> Thank you for your review, we appreciate the thorough feedback. Let us address your comments in detail.
>
> > However, the evaluation is performed only on the CIFAR dataset. It would be better if the authors could also include other commonly used benchmarks for image classification.
>
> In general, we agree that it would be good to include more benchmarks, albeit this was not the main focus of our work. Scaling vanishing ideal algorithms to larger datasets remains an open problem and is an active area of research [Wirth et al., 2023], our work is a step in this direction, with multiple contributions to that end. However, to directly address your concern, we have run preliminary experiments on TinyImageNet using ResNet-50. We observe that the performance is comparable to the CIFAR-100 and CIFAR-10 experiments, indicating that the method works for larger datasets. We will add these results to the paper and run further experiments on image classification benchmarks for the camera-ready version of the paper. Please find the results here https://imgur.com/a/Beh0DN7.
>
>
> > The only issue is that there seem to be no direct comparison between VI-Net and the base model. For example, under what condition does , i.e. VI-Net has a lower spectral complexity?
>
> The main motivation for using the spectral complexity framework is exactly that it allows for comparison of generalization bounds of different neural network architectures and does more than simply providing a learning guarantee. We can employ Theorem 4.5 to derive for each layer $L'$ a bound on the norms of the newly introduced parameters that is strictly lower than that of the baseline network. This provides a theoretical justification for the use of VI-Net.
>
> Example: suppose we remove the last 3 layers in a pretrained NN. We can compute the spectral norms of their weight matrices $W_1, W_2, W_3$ and compute the product of these terms, i.e., $\prod_{i = 1}^3 \|W_i\|_2$.
>
> Similarly, we can compute $\sum_{i = 1}^3\|W_i^T\|_{2,1}^{-2/3}\|W_i\|_2^{-2/3}$ explicitly. Next, we can *choose* the number of monomial terms $S$, the number of polynomials $N$, the highest degree $d$, the bound on the norm of the coefficient vectors of the polynomials $\tau$ (Note that these are all hyperparamters of our algorithm pipeline).
>
> If we have further bounds $\lambda_1, \lambda_2$ (cf. Theorem 4.5) on the norms of the final linear weight matrix (either by explicit constraints or implicit regularization), then we can always ensure that
> - $2^dd\tau \lambda_1 \sqrt{NS} \leq \prod_{i = 1}^3 \|W_i\|_2$
>
> and
>
> - $2^{2d/3}S^{2/3}+ N^{2/3}S^{1/3} + \lambda_2^{2/3} \leq \sum_{i = 1}^3\|W_i^T\|_{2,1}^{-2/3}\|W_i\|_2^{-2/3}$.
>
> Thus, the spectral complexity of VI-Net is lower compared to the original baseline NN by Theorem 4.5.
>
> > The codebase in the supplementary material seems not runnable.
>
> We apologize, this was an oversight on our side while anonymizing the code. We have now ensured that all necessary files are included in the repository, and you should be able to run the code without any problems. If the Area Chair permits, we would be happy to share the complete codebase with you (currently only links to images are allowed).
>
>
> > Also, a dedicated section to related works would be helpful.
>
> We are fairly confident to have included all relevant works with respect to vanishing ideal algorithms, but we appreciate the suggestion of adding a dedicated related work section. We think this will improve the presentation of our work and will add this section to the revision of the paper as soon as possible. Thank you for your suggestion.
>
> > Can we apply the methodology of VI-Net to regression tasks?
>
> This is an interesting idea. Indeed, one can apply the standard approach of discretizing the regression problem transforming it into a classification problem (i.e. binning numerical outputs). Do you have any other suggestions? We believe this could be an interesting application and thank the reviewer for this idea.
>
> > Moreover, apart from building new models for prediction as you have done in this paper, can we use vanishing ideal algorithms to analyze the structure of the latent manifold itself and provide some kind of interpretability?
>
> We do observe a high correlation between the intrinsic dimensionality of the data and the complexity of the constructed generators (cf. Figure 12). This aligns well with the findings in the broader literature and we believe that this provides insights about the manifold, although relatively hard to interpret.
>
> > L201, 202, 304, 285: [...]
>
> Thank you, we fixed the typos and notational issues.
>
> > L323: What does $s \in \mathcal{O}(S)$  mean?
>
> The size of the intermediate weight matrix scales linearly with the number of monomials.
>
> > L392: Does elapsed time in Table 1 refer to the running time of the vanishing ideal algorithm?
>
> Yes.
>
> Thanks again for your review, we hope to have addressed your concerns and questions. If you have any further remarks, please let us know.

---

> > ### Comment · Reviewer_9he7 · 2025-04-03
> >
> > Thank you for the response. I will keep my score.

---

### Official Review · Reviewer_cAHx · 2025-03-12

**Overall Recommendation:** 1

**Summary:**

This paper introduces efficient methods for approximating vanishing ideals of class "manifolds" in a latent space of a neural network classifier, and uses the resulting approximate vanishing ideals to replace later layers of a truncated classifier neural network with a linear combination of polynomial features, calling the resulting classifiers "VI Nets". The paper includes theoretical discussion suggesting that such VI Nets might enjoy improved generalization properties, and demonstrates empirically that for a given transition layer, they preserve more accuracy from the initial classifier network than simply replacing the later layers with a linear head. The paper shows that VI Nets can navigate an accuracy-throughput tradeoff.

# Update after rebuttal

I appreciate the authors' thorough response to my questions and comments. While they addressed a number of technical concerns, I maintain my reject recommendation.

**Claims And Evidence:**

- "Such manifolds can often be described as the zero set of a system of polynomial equations" hinges on what is meant by often. I agree that many standard examples of manifolds are zero sets of polynomials, however in some theoretical senses "almost all" manifolds fail to be algebraic. See e.g. https://mathoverflow.net/questions/4895/the-relationship-between-complex-and-algebraic-geomety
- Similarly this is false: "A key observation, and central motivation for our work, is that if the latent space is a manifold, it can be algebraically characterized as the zero set of certain polynomials". In general, the zero set of the vanishing ideal will be a super set of the original manifold. There are similar issues elsewhere, such as "Unlike methods that approximate manifolds through indirect means like simplicial complexes (Khrulkov & Oseledets, 2018) or k-nearest neighbor graphs (Tsitsulin et al., 2020), our approach directly captures their algebraic structure through polynomial roots"
- "A key insight is that data points from different classes in the latent space, while not linearly separable, can be distinguished through polynomial separability" should say something like "while not necessarily linearly separable." Since they might be linearly separable, and in some theoretical frameworks asymptotically *are* linearly separable (for example, in the infinite width limit).
- the last paragraph of section 2.1 effectively assumes that the $Z^k$ are pairwise disjoint. This is probably always true in practice, but it should be stated as an assumption. On the topic of this section, squares would be preferable to absolute values since they are polynomials :).

**Essential References Not Discussed:**

Not beyond those linked above and below.

**Experimental Designs Or Analyses:**

Significant concern: I do not understand why an accuracy difference is observed in figure 3, even with 0% of the layers removed. In particular, I do not understand why the Linear Head method drops all the way to 65% accuracy, which is quite bad for a resnet34 on CIFAR 100. There are publicly available training scripts that claim to achieve accuracy in the high 70s for resnet34s on CIFAR 100.

Regarding table 2: I think that including a resnet18 on CIFAR100 would be a useful baseline. More broadly, there are many methods which practitioners can use to achieve throughput improvements of the scale being demonstrated by VI nets (up to about 2x), with minimal impact on accuracy (hardware friendly inference frameworks, quantization ...). Additional comparison with/discussion of these would help contextualize table 2.

**Methods And Evaluation Criteria:**

I think the limitations section could mention something about experiments only covering ResNets and CIFAR datasets (in particular this restricts to the vision domain).

The experiments primarily focus on "end to end testing" of the entire VI Net package. I could imagine additional experiments that would more directly target the underlying hypotheses (e.g. empirically computing some of the terms appearing in the spectral complexity estimates ...)

**Other Comments Or Suggestions:**

- Def. 4.2, should it be "d choose i"?
- Lem. 4.4, what are the subscripted $d_i$?
- " Specifically, if the product of the spectral norms of the truncated layers is smaller than the bound for the newly added layers ..." Do we not want the opposite to ensure the VI Net has lower spectral complexity? Regardless, using some notation here would make the sentence easier to parse.
- it would be helpful to have a companion to figure 3 that incorporates the parameter counts of the various models.
- In table 1 is elapsed time the time it took to compute (approximate) ideal generators?

**Other Strengths And Weaknesses:**

My primary concerns with this paper:

- there's a variety of work showing that classes become increasingly linearly separable in later and later neural networks layers (https://arxiv.org/abs/1610.01644, https://arxiv.org/abs/2004.06093 to name a couple).
- In particular, the former paper shows that truncating a neural network at an intermediate layer, and simply immediately learning a linear layer can result in performance comparable to that of the original network, *without* any intermediate steps of dimension reduction, tanh rescaling and bit crushing, vanishing ideal computation ... NOTE: I realize this baseline appears in Fig. 3, and that the proposed method of VI nets is shown to result in higher accuracy, however the fact that linear probes are not mentioned until the penultimate page is an issue for me, since as a reader, I was thinking "what about linear probes" on page 1.
- The latter reference suggests that the process of neural network training is already encouraging class separation, one extreme example of this being "neural collapse" (https://arxiv.org/abs/2008.08186, https://proceedings.mlr.press/v145/e22b.html). So another baseline if one wanted a shallow or network that achieves linear class separation would be to try to train a shallow or network to the point of neural collapse. I do not see this baseline in the paper.
- I could be mistaken, but the baselines suggested above might be amenable to similar spectral complexity analysis (definitely the first one is, since it will share the feature that lots of terms coming from the common first L layers cancel).

**Questions For Authors:**

Not beyond those above.

**Relation To Broader Scientific Literature:**

Things that come to mind are: empirical science of representations of data in neural network latent space, and application of computational algebraic, geometry to data science. These are both somewhat broad areas, I don't have specific references in mind.

**Theoretical Claims:**

I did not read the proofs in the appendix. Regarding the theoretical claims, while the paper derives formulas that could be used to estimate spectral complexity of VI Nets (and relate their complexity to that of the network from which they were derived), since there are no theorems specifically proving that they have lower spectral complexity under certain assumptions, I think the introduction of the paper should be more cautious about suggesting improved generalization properties of VI Nets.

---

> ### Author Rebuttal · Authors · 2025-04-01
>
> Thank you for your thorough and insightful comments. We have grouped related concerns and clarifications below and revised our paper accordingly.
>
> **Overall Clarifications and Algebraic Geometry Context**
>
> We used the manifold hypothesis to motivate our work, but we agree that currently, some of the wordings are not entirely mathematically precise. Let us make a few clarifications:
>
> > in some theoretical senses "almost all" manifolds fail to be algebraic
>
> In line 42, we cited [Nash, 1952] to reference Tognoli's Theorem that every *compact smooth* manifold is *diffeomorphic* to a nonsingular algebraic set. We revised the wording to clarify that some manifolds may not be described by polynomials.
>
> > the zero set of the vanishing ideal will be super set of the original manifold.
>
> Please note that we in lines 162-167 we explicitly state that "recovering [the underlying manifold] $U_k$ exactly from a finite sample is impossible without structural information, as $U_k$ could range from being as minimal as $Z_k$ to as broad as $\mathbb{R}^n$" and provide further context in section 2.3.
>
> > "while not necessarily linearly separable."
>
> We agree and will modify the statement accordingly. Thanks!
>
> > the last paragraph of section 2.1 effectively assumes that the $Z_k$ are pairwise disjoint
>
> Correct, we implicitly state this in line 80-82, but will make it more explicit here.
>
> **Experimental Evaluation and Baselines**
>
> *Linear Probing:*
>
> First of all, we agree that Linear Probing should be mentioned earlier in the paper and will revise the manuscript to make this more clear. Thank you for pointing out that this is not easily accessible. Secondly, we understand the confusion regarding the accuracy drop in Fig. 3, whose content was poorly communicated from our side. At 0% removed, we removed *0% of convolutional layers*, but still remove the avg.-pooling and linear classifier (lines 332-334, despite badly worded), which may result in a performance drop. We make this more clear to not leave the reader behind to why "0% removal" actually does something. Last but not least, we retuned the hyperparams of the linear probing baseline, improving its performance (https://imgur.com/a/SOpcrEJ). In general, the accuracy drop aligns with the paper you mentioned, reporting a 6% performance drop when using a linear head after the last convolution - which is a non-insignificant drop (cf. Figure 4 in arxiv.org/abs/1610.01644). Now, while VI-Net still outperforms, both methods perform similarly when removing the trailing convolutional layers.
>
> We similarly observe the phenomenon that later layers lead to simpler, sometimes linear, polynomials (cf. Figure 11), aligning with the finding that classes become increasingly linearly separable in later layers; the neural collapse phenomenon is further supported by the good performance of the linear probes. VI-Nets could be seen as a natural extension of linear probing. Could you elaborate on what kind of additional baseline you would like to see?
>
>
> *Further experiments:*
>
> To broaden our experimental evaluation, we added TinyImageNet experiments (https://imgur.com/a/Beh0DN7). Further, we conducted additional experiments measuring excess risk to support our hypothesis (please cf. our response to reviewer Gw8X).
>
> We will add the evaluation of ResNet-18 on CIFAR-100 to properly contextualize the throughput improvements, however noting the following: While VI-Net's speedup might not match quantization and hardware optimization, we still observed a notable speedup despite non-optimized hardware, indicating the usefulness of the found generators in describing data manifolds.
>
> **Remarks regarding theory**
>
> We do agree that not every VI-Net has a lower spectral complexity. Theorem 4.5 allows to choose the hyperparameters of our pipeline such that the bound we have on the newly added terms is lower than what can be measured from the baseline NN (see the answer we provided to reviewer 9he7 on the question on when VI-Net has a lower spectral complexity).
>
> Further, you are correct that the baselines can be analyzed using the same spectral complexity framework. This enables us to compare the spectral complexities of the baseline NN, VI-Net, and linear probing or a shallow wide NN within a unified framework. We think this is a strength of our work.
>
> **Other remarks**
>
> > Fig 12: the number of generators and the dimension of the variety are anti-correlated.
>
> We think this is a result of the *approximate* VI algorithms. If the underlying data manifold has lower dimension (e.g. a line), fewer monomials are needed to construct polynomials that approximately approximately vanish on the sample.
>
> * We fixed the typos and will include your suggestions, thanks.
> * Lem. 4.4: The $d_i$ are the degrees of the InEx activation functions.
> * Yes, in Tab. 1 that is the elapsed time to compute the generators.
>
> We hope to have addressed your concerns, please let us know if further clarification is needed.

---

### Official Review · Reviewer_Gw8X · 2025-03-13

**Overall Recommendation:** 3

**Summary:**

Present a method to analyse latent manifolds - object representations at intermediate layers of deep networks - by finding a small set of polynomials which zero out at one object but not at other objects. This effectively trunfaces the deep network and replaces the head with a single, simple, non-linear layer - with each feature stemming from a single polynomial of a single object. Such construction is beneficial twofold: in demonstrating a parameter-efficient network with comparable classification performance as the original network and in characterising the latent manifold.
This construction pulls several different tricks to get those "vanishing" polynomials - dimension reduction by PCA, sparse ideal learning algorithms pruning approaches to allow for efficient inference, reduced precision implementation and numerical rescaling (if I could count them all).

**Claims And Evidence:**

Most claims seem to be well-supported by evidence, with the numerical experiments using ResNet on CIFAR-10/100 used to demonstrate the applicability to real-world data.

The theoretical analysis is an exception; while I do not doubt the validity of the analysis (demonstrating reduced "spectral complexity"), I have serious doubts about the implications. Those "learning guarantees" are usually vavous bounds, and thus, an improved upper bound on something may or may not lead to improved performance. As far as I could see, such improved performance was NOT demonstrated here.

**Essential References Not Discussed:**

As mentioned, this manuscript makes novel contributions to a well-researched area and thus naturally lack some "essential references", e.g.,:

 * Separability and geometry of object manifolds in deep neural networks.
 * Do Wide and Deep Networks Learn the Same Things? Uncovering How Neural Network Representations Vary with Width and Depth
 * Deep neural networks rival the representation of primate IT cortex for core visual object recognition

**Experimental Designs Or Analyses:**

Yes.

**Methods And Evaluation Criteria:**

Yes, the evaluation criteria look good to me. See some suggestions/improvements below.

**Other Comments Or Suggestions:**

The results are constructed such that the number of monomial terms is a product of the algorithm, while the maximal degree and representation dimension are fixed. I think this is a mistake - the correct comparison should have been:
 * Using a fixed monomials budget (e.g., maximal degree of d=5, 128 dimensions, and the best 100 (or 200 or 1000) monomials), what is the resulting accuracy?
 * Aiming for a fixed target accuracy (e.g., 90%), how many monomials are needed at each layer?

The above analysis would have demonstrated how and if later layers of deep network have superior representations so they need a smaller number of monomials for a given accuracy level or better accuracy for a given monomials budget.

Unrelated, the presentation of "Percentage of Layers Removed" seems unnatural to me: I would expect to look at "percent of layers used" with increasing accurace.

**Other Strengths And Weaknesses:**

Strengths:
 * The manuscript presents a novel idea of truncating a trained network and creating a new kind of learned, object-focused final layer.
 * A large collection of technical solutions is needed to bring this idea close to real-world applicability.
 * The very interesting results demonstrate both that this endeavour can be successful and that it may be useful to gain insight into the structure of latent representations.

Weaknesses:
 * Not enough is done to build on this analysis to say something on the structure of the latent manifolds. See one suggestion below, perhaps for a future manuscript.

**Questions For Authors:**

* Can you provide some analysis on the effect of the maximal degree (keeping the number of monomials fixed)?
 * Are you certain you need to "limit the maximal degree to avoid overfitting"? Assuming the number of monomials is fixed, I'm not sure about that.
 * Can you analyse the accuracy using a fixed number of monomials constructed from different layers, or the number of monomials needed to achieve 90% accuracy?
* Can you demonstrate a clear benefit of the fact that "the spectral norm of the VI-Net is provably lower than that of the base NN"?
* Can you clarify which tricks (e.g., lower precision aritmatic) are essential for the results and which just improve your running time slightly?

**Relation To Broader Scientific Literature:**

This is a novel approach to handling a well-researched area: understanding object representations in deep networks. The novel ideas presented show how to solve many practical problems (finding ideals is seems at first irrelevant on computational grounds).

**Theoretical Claims:**

No, I do not have the capacity to verify the correctness of the theoretical claims.

---

> ### Author Rebuttal · Authors · 2025-04-01
>
> Thank you for your thoughtful review and for acknowledging the contributions of our work.
>
> > Those "learning guarantees" are usually vavous bounds, and thus, an improved upper bound on something may or may not lead to improved performance.
>
> We agree that these bounds are often vacuous. In that sense, we agree that improved upper bounds do not necessarily imply improved performance; thank you for that remark, we will make this more clear in the paper. Our main theoretical contribution is embedding VI-Nets into the established framework of Spectral complexity, with our main theorem providing a theoretical basis for selecting hyperparameters.
>
> > As mentioned, this manuscript makes novel contributions to a well-researched area and thus naturally lack some "essential references"
>
> Thank you for providing these references. We believe the provide valuable context and will add them to the paper.
>
> > to say something on the structure of the latent manifolds. See one suggestion below, perhaps for a future manuscript.
>
> We also believe this is an interesting future direction of our work. Having constructed these polynomials, we are now able to use the entire toolbox of computational algebra to analyse our found structures. We do believe our work is already quite dense and leave these useful applications to future work - we will address the individual suggestions below!
>
> > * Using a fixed monomials budget what is the resulting accuracy?
> > * Aiming for a fixed target accuracy (e.g., 90%), how many monomials are needed at each layer?
>
> We ran experiments on CIFAR100, testing monomial counts from 50, 100, and increasing by 100, noting their maximal degree. We limit ourselves to the following sparsities, as higher sparsity results in reaching less than 70% accuracy.
>
> | % of Layers Removed | Monomial Count | Max Degree among best Monomials |
> |-|-|-|
> | 7.1%|50| 1 |
> | 12.2%|50| 1 |
> | 17.5%|300| 2|
> | 25.1%|500| 2|
> | 30.4%|200| 2|
> | 33.3%|300| 2|
> | 38.4%|600| 2|
> | 43.9%|900| 3|
>
> We observe that deeper layers need fewer and lower-degree monomials. Deeper layers achieve higher accuracy with a fixed monomial count, but adding more monomials eventually stops improving performance.
>
> | # Monomials | 17.5% layers removed | 33.3% layers removed | 75.6% layers removed|
> |-|-|-|-|
> | 50          | 68.47%  | 59.42%  | 43.42%  |
> | 100| 68.31%  | 59.83%  | 45.63%  |
> | 200| 69.66%  | 65.43%  | 48.63%  |
> | 400| 71.84%  | 72.35%  | 47.80%  |
> | 600| 72.54%  | 72.65%  | 47.26%  |
> | 800| 72.90%  | 72.73%  | 47.46%  |
> | 1000        | 72.96%  | 72.91%  | 48.81%  |
>
>
> > Can you provide some analysis on the effect of the maximal degree (keeping the number of monomials fixed)? Are you certain you need to "limit the maximal degree to avoid overfitting"? Assuming the number of monomials is fixed, I'm not sure about that.
>
> Given that tanh-rescaling ensures that the output lies in the unit hypercube, a high degree polynomial with a fixed number of monomials could overfit/vanish on the dataset. However, fixing the number of monomials also has a regularizing effect (cf. Tables below). Our learning guarantee in Theorem 4.5 includes both the highest degree $d$ and the number of monomials $S$, indicating both are relevant. But we agree that our statement is a bit vague, in practice we chose $d = 5$ as maximal degree mainly because we observed that the algorithms never constructed polynomials of degree higher than $5$.
>
> > Can you demonstrate a clear benefit of the fact that "the spectral norm of the VI-Net is provably lower than that of the base NN"?
>
> The lower spectral norm of the VI-Net means it might generalize better, resulting in a **smaller difference between training and testing errors**. For example, below, we see that the VI-Net has a smaller gap between train and test errors in almost all configurations compared to the base neural network.
>
> | % of Layers Removed | Gap Train vs Test Error     | Δ from Baseline |
> |-|-|-|
> | 75.8%| 1.7843| +0.6643|
> | 50.1%| 0.9639| −0.1561|
> | 43.9%| 0.8502| −0.2698|
> | 25.1%| 0.9005| −0.2195|
> | 7.1%| 1.1024| −0.0176|
>
> **Baseline (CIFAR-100, ResNet-34)** = 1.12
>
> For the experiment you suggested, one can analyse the impact of the number of monomials $S$ on the generalization gap. Fewer monomials typically lead to a smaller generalization gap.
>
> | # Monomials | 17.5% removed | 33.3% removed | 75.6% removed |
> |-|-|-|-|
> | 50| 20.27%| 11.10%| 0.87% |
> | 100| 20.26%| 12.04%| 0.67% |
> | 200| 26.64%| 19.80%| 4.45% |
> | 400| 27.55%| 25.67%| 11.88%|
> | 600| 27.17%| 26.08%| 18.18%|
> | 800| 26.94%| 26.31%| 24.79%|
> | 1000| 26.91%| 26.27%| 28.68%|
>
> **Baseline (CIFAR-100, ResNet-34)** = 25.25
>
> > Can you clarify which tricks are essential for the results and which just improve your running time slightly?
>
> Tanh-rescaling ensures convergence. Stochastic algorithms speed up computations the most, while lower precision arithmetic offers mainly memory benefits.
>
> We hope to have addressed your concerns. If you have any further questions, please let us know.

---

> > ### Comment · Reviewer_Gw8X · 2025-04-08
> >
> > I have read the authors' rebuttal and other reviewers' comments and would keep my current (high) score. Interestingly, I probably agree with most of cAHx's comments (leading them to reject), but still believe this is an interesting the valuable venue which complements the linear separability literature, while still being somewhat shadowed by its success.

---

### Decision · Program_Chairs · 2025-05-01

**Decision:**

Accept (poster)

**Comment:**

The article presents methods for approximating vanishing ideals in latent space of neural nets and uses this to identify classifiers with fewer layers.

One reviewer recommends strong accept, indicating that the manuscript presents a novel idea and interesting results that may be useful to gain insight. At the same time they indicate that not enough is done to build on this analysis to say something about the structure of the latent manifold and that they do not have the capacity to verify the correctness of the theoretical claims. The discussion indicates that the reviewer agrees with comments from another (critical) reviewer, but that they believe this is an interesting contribution nonetheless.
One reviewer recommends reject with concerns that persisted after the rebuttal. Concerns pertain to the lack of theoretical results establishing differences in complexity as well as certain claims. Some of the notation and presentation aspects were clarified in the rebuttal. Authors conceded that not every VI Net has lower spectral complexity.
Another reviewer recommends weak accept indicating that the experiments are comprehensive and demonstrate effectiveness, but also that the article mainly applies existing algorithms. They describe the proposed method as straightforward and sensible for improving parameter efficiency. They maintained their score after the rebuttal.
Another reviewer recommends weak accept, indicating that the authors investigate generalisation based on existing results regarding spectral complexity.

I find that the article, albeit not being strong in all regards, contributes neat ideas that could generate further discussions leading to more insights. Given that referees overall lean towards accept, I recommend accept. I ask hat the authors take the reviews and rebuttal discussion carefully into consideration when preparing the final version of the manuscript.